



# Development and Evaluation of the Aerosol Forecast Member in

# NCEP's Global Ensemble Forecast System (GEFS-Aerosols v1)

Li Zhang[1,2], Raffaele Montuoro[1,2], Stuart A. McKeen[1,3], Barry Baker[4,5], Partha S. Bhattacharjee[6], Georg A. Grell[2], Judy Henderson[2], Li Pan[6], Gregory J. Frost[3], Jeff McQueen[7], Rick Saylor[8], Haiqin Li[1,2], Ravan Ahmadov[1,2], Jun Wang[7], Ivanka Stajner[7], Shobha Kondragunta[9], Xiaoyang Zhang[10], Fangjun Li[10]

[1]CIRES, University of Colorado, Boulder, CO, US;
[2]Global Systems Laboratory, Earth System Research Laboratory, NOAA, Boulder, CO, US;
[3]Chemical Sciences Laboratory, Earth System Research Laboratory, NOAA, Boulder, CO, US;
[4]NOAA Air Resources Laboratory, College Park, MD, US;
[5]Cooperative Institute for Climate and Satellites, University of Maryland, College Park, MD, US;
[6]I.M. Systems Group at NCEP/NWS/EMC, College Park, MD, US;
[7]Environmental Modeling Center, National Weather Service, Greenbelt, MD, US;
[8]NOAA Air Resources Laboratory, Oak Ridge, TN, US;
[9]NOAA/NESDIS Center for Satellite Applications and Research, Greenbelt, MD, US;
[10]Department of Geography, South Dakota State University, Brookings, SD, US

*Correspondence to: Li Zhang (kate.zhang@noaa.gov)

CIRES, University of Colorado Boulder

GSL EPAD, NOAA ESRL

325 Broadway David Skaggs Research Center R/GSL1

Boulder, CO 80305

1-303-497-3956



**Abstract.**
NOAA's National Weather Service (NWS) is on its way to deploy various operational prediction applications using the Unified
Forecast System (https://ufscommunity.org/), a community-based coupled, comprehensive Earth modeling system. An aerosol
model component developed in a collaboration between the Global Systems Laboratory, Chemical Science Laboratory, the
Air Resources Laboratory, and Environmental Modeling Center (GSL, CSL, ARL, EMC) was coupled online with the
FV3 Global Forecast System (FV3GFS) using the National Unified Operational Prediction Capability (NUOPC)-based NOAA
Environmental Modeling System (NEMS) software framework. This aerosol prediction system replaced the NEMS GFS
Aerosol Component (NGAC) system in the National Center for Environment Prediction (NCEP) production suite in September
2020 as one of the ensemble members of the Global Ensemble Forecast System (GEFS), dubbed GEFS-Aerosols v1. The
aerosol component of atmospheric composition in GEFS is based on the Weather Research and Forecasting model (WRF-
Chem) that was previously included into FIM-Chem (Zhang et al, 2021). GEFS-Aerosols includes bulk modules from the
Goddard Chemistry Aerosol Radiation and Transport model (GOCART). Additionally, the biomass burning plume rise module
from High-Resolution Rapid Refresh (HRRR)-Smoke was implemented; the GOCART dust scheme was replaced by the
FENGSHA dust scheme (developed by ARL); the Blended Global Biomass Burning Emissions Product (GBBEPx V3)
provides biomass burning emission and Fire Radiative Power (FRP) data; and the global anthropogenic emission inventories
are derived from the Community Emissions Data System (CEDS). All sub-grid scale transport and deposition is handled inside
the atmospheric physics routines, which required consistent implementation of positive definite tracer transport and wet
scavenging in the physics parameterizations used by NCEP's operational Global Forecast System based on FV3 (FV3GFS).
This paper describes the details of GEFS-Aerosols model development and evaluation of real-time and retrospective runs using
different observations from in situ measurement, satellite and aircraft data. GEFS-Aerosols predictions demonstrate substantial
improvements for both composition and variability of aerosol distributions over those from the former operational NGAC
system.



## 1. Introduction

The operational air quality predictions in National Oceanic and Atmospheric Administration (NOAA)'s National Weather Service (NWS) contribute to the protection of lives and health in the US (https://airquality.weather.gov). These predictions are used by state and local air quality forecasters to issue official air quality forecasts for their respective areas. The U.S. Environmental Protection Agency (EPA) and the Centers for Disease Control and Prevention (CDC) also use the NOAA forecasts for applications with wildfire, health and smoke vulnerability assessments. Exposure to fine particulate matter, i.e., aerosol particles with diameters of 2.5 μm and smaller ($PM_{2.5}$), is recognized as a major health concern and the associated mortality rate is estimated to be higher than the five specific causes of death examined by the global burden of disease (GBD, Burnett et al., 2018).

The role of aerosols in Numerical Weather Prediction (NWP), through interaction with atmospheric radiation and precipitation physics (direct, semidirect, and indirect effect), and their impact on meteorological fields in both weather and climate scale, have been widely recognized in many studies [e.g. Fast et al. 2006, Levin and Cotton, 2009; Chen et al., 2011; Grell et al. 2011; Forkel et al. 2012; Muhlbauer et al., 2013; Xie et al., 2013; Yang et al., 2014; Wang et al., 2014a, 2014b; Colarco et al., 2014]. For example, global and regional models established a connection between dust emissions and weather patterns over synoptic-to-seasonal time scales [Haustein et al., 2012; Zhao et al., 2010]. Results from NASA's Goddard Earth Observing System (GEOS-5) forecasting system showed that the net impact of the interactive aerosol associated with a strong Saharan dust outbreak resulted in a temperature enhancement at the lofted dust level and a reduction near the surface levels, which improved forecasts of the African easterly jet (AEJ) [Reale et al., 2011]. Similar improvements in NWP were reported by Tompkins et al. [2005] who found that an updated dust climatology led to a northward shift of the AEJ in the European Centre for Medium-Range Weather Forecasts (ECMWF) NWP model. Wang et al [2014c] revealed that anthropogenic aerosols from Asian pollution enhanced precipitation and poleward heat transport, resulting in intensifying the Pacific storms. Toll et al. [2015] showed considerable improvement in forecasts of near-surface conditions during Russian wildfires in the summer of 2010 by including the direct radiative effect of realistic aerosol distributions. Furthermore, the microphysical and thermodynamic effects from aerosols may play a role in impacting development of tropical cyclones [Rosenfeld et al. 2012]. Additional studies at operational weather centers indicate the importance of including aerosol feedback in NWP for operational forecasting. The inclusion of the direct and indirect effects of aerosols in the global NWP configuration of the Met Office Unified Model (Met UM) indicated that using prognostic aerosols led to better temporal and spatial variations of atmospheric aerosol optical depth (AOD) and was of particular importance in cases of large sporadic aerosol events such as large dust storms or forest fires [Mulcahy et al., 2014]. At ECMWF, Rodwell and Jung [2008] showed an improvement in forecast skill and general circulation patterns in the tropics and extra-tropics by using a monthly aerosol climatology rather than a fixed climatology in the ECMWF global forecasting system. Later on, ECMWF replaced the aerosol climatologies with aerosols from a reanalysis of atmospheric composition produced by the Copernicus Atmosphere Monitoring Service [Bozzo et al, 2020]. At National Centers for Environmental Prediction (NCEP), the operational RAPid refresh (RAP) and High-Resolution



Rapid Refresh (HRRR) storm scale modeling systems now include the impact of aerosols from biomass burning emissions on
radiation.
NCEP, in collaboration with the NASA/Goddard Space Flight Center (GSFC), developed the NEMS Global Forecast System
(GFS) Aerosol Component version 1 (NGACv1) for predicting the distribution of global atmospheric aerosols [Lu et al., 2010].
NGACv1 was implemented in 2012 and provided the first operational global dust aerosol forecasting capability at NCEP [Lu
et al., 2016]. In NGACv1 an in-line aerosol module based on the Goddard Chemistry Aerosol Radiation and Transport
(GOCART) model from GEOS-5 [Chin et al., 2000], but limited to dust only, was used. The NGACv1 used the Earth System
Modeling Framework (ESMF) to couple the aerosol module with the GFS. Later on, NCEP implemented a multispecies aerosol
forecast capability NGACv2, based on NGACv1 through collaborations among NCEP, NASA/GSFC, the NESDIS Center for
Satellite Applications and Research (STAR), and the State University of New York at Albany [Wang et al., 2018]. NGACv2
included additional aerosol species of sea salt, sulfate, organic carbon, and black carbon from the updated GOCART modules,
and biomass burning emissions from the NESDIS STAR's Global Biomass Burning Product (GBBEPx) as well as GSFC's
Quick Fire Emission Data Version 2 from a polar-orbiting sensor (QFED2). Both science and software upgrades in the global
forecast system were updated and implemented into NGACv2 in March 2017 to provide 5-day multispecies aerosols forecast
products at the T126 L64 resolution, approximately 100 km.
In July 2016, NOAA took a significant step toward developing a state-of-the-art global weather forecasting model by
announcing the selection of a new dynamic core developed at NOAA Geophysical Fluid Dynamics Laboratory (GFDL) to
upgrade the GFS. The GFDL Finite-Volume Cubed-Sphere Dynamical Core (FV3) replaced the spectral GFS core in June of
2019 to drive global NWP systems with improved forecasts of severe weather, winter storms, and tropical cyclone intensity
and track. NOAA is now on the way to integrate various operational applications into the Unified Forecast System (UFS), a
comprehensive, community-based coupled Earth modeling system, designed as both a research tool and the basis for NOAA
operational forecasting applications.
Here we describe a new aerosol model component developed through collaborative efforts among the Global Systems
Laboratory (GSL), the Chemical Science Laboratory (CSL), and the Air Resources Laboratory (ARL), Environmental Model
center (EMC) and STAR. This aerosol component was implemented operationally on September 2020 to provide five-day
global aerosol forecasts as one member of the Global Ensemble Forecast System (GEFS): GEFS-Aerosols v1. The aerosol
component is designed as an independent model component for the NOAA Environmental Modeling System (NEMS)
framework, and includes a coupling interface based on the National Unified Operational Prediction Capability (NUOPC) Layer
for model interoperability. All chemistry, aerosol, and emission modeling processes reside and run within this model
component. GEFS-Aerosols shows a substantial improvement for both composition and variability of aerosol distributions
over those from the previous global aerosol prediction system NGAC. The model predicted global aerosol products from
GEFS-Aerosols are also used for other applications, such as to provide lateral boundary conditions for NOAA's regional
National Air Quality Forecast Capability (NAQFC), satellite sea surface temperature (SST) physical retrievals, and the global
solar insolation estimation [Wang et al., 2018].



The current study presents the development of GEFS-Aerosols and evaluations of its performance in real time and retrospective
experiments. Section 2 describes the coupling components of the GEFS-Aerosols member, including the atmospheric
component of FV3GFS model, the aerosol component, and the observation, reanalysis, and model data used for evaluation
and comparison. The emission inventories of both anthropogenic emission and biomass burning emissions and other chemical
input data are presented in Section 3. Section 4 and Section 5 are the evaluations of Day-1 real-time forecast since July 2019
and the Day-1 retrospective forecast for the ATom-1 periods of 2016 summer, respectively. The conclusions and future plans
are summarized in Section 6.

## 2. Model and data

### 2.1 Descriptions of GEFS-Aerosols

#### 2.1.1 FV3GFS

The global Finite-Volume cubed-sphere dynamical core (FV3) developed by GFDL was chosen by NOAA as the non-
hydrostatic dynamical core to be the Next Generation Global Prediction System (NGGPS) of the National Weather Service in
the US. Currently the FV3 was successfully implemented within the GFS, and the FV3-based GFSv15 became operational on
June 2019, providing the metrological basis for coupling with a simple aerosol prediction component. All sub-grid scale
transport and deposition is handled inside the atmospheric physics routines of simplified Arakawa–Schubert (SAS) scheme,
which required consistent implementation of positive definite tracer transport and wet scavenging in the physics
parameterizations in GFSv15 and subsequent GEFSv12.

#### 2.1.2 Aerosol component

The current atmospheric composition option in the GEFS-Aerosols model is based on the simple bulk aerosol modules from
WRF-Chem [Grell et al., 2005; Powers et al., 2017], and were previously used in the global Flow-following finite-volume
Icosahedral Model (FIM), as FIM-Chem [Zhang et al, 2021], including aerosol modules from GOCART. The GOCART
aerosol modules use simplified sulfur chemistry for sulfate simulation, bulk aerosols of black carbon (BC), organic carbon
(OC), and sectional dust and sea salt [Chin et al., 2000]. For OC and BC, the hydrophilic and hydrophobic components are
considered and the chemical reactions for gaseous sulfur oxidations are calculated using prescribed OH, $H_2O_2$, and $NO_3$ fields
for gaseous sulfur oxidations [Chin et al., 2000]. The marine dimethyl sulfide emissions from Lana et al. [2011] are used with
monthly averaged values. Recently, some modifications and updates have been implemented, including the biomass burning
plume rise module adapted from WRF-Chem, the capabilities of using the biomass-burning emission calculations based on
the Blended Global Biomass Burning Emissions Product [GBBEPx V3, Zhang et al., 2014] emission and Fire Radiative Power
data provided by NESDIS as well as the application of the global anthropogenic emission inventories from Community
Emissions Data System (CEDS).

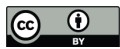



The sea salt scheme was updated to the most recent version with five size bins based on NASA's 2nd-generation GOCART
model.
A new dust emission scheme, referred to as FENGSHA, was implemented in GEFS-Aerosols. The scheme, which is also used
in NOAA's National Air Quality Forecast Capability, is modified from the original Owen equation [Tong et al., 2017, Owen,
1964; Shao et al., 1993],
$$F = \sum_{j=1}^{N} K \times A \times \frac{\rho}{g} \times S \times u_* \times \left( u_*^2 - u_{*tj}^2 \right) \quad \text{for u* > u*t}_j \quad (1)$$
where $N$ is the number of soil types in a particular grid cell, $K$ is the ratio of vertical to horizontal emission flux, $A$ represents
particle supply limitation (availability), $\rho$ is air density, $g$ is gravitational acceleration, $S$ is the soil erodibility potential, $u*$ is
friction velocity, and $u*_{tj}$ is the threshold friction velocity for soil type $j$ [Shao et al., 1993]. Dust emission is calculated only
when friction velocity exceeds the designated threshold value for the land use type and soil texture.
What makes FENGSHA unique is the way in which the threshold values are determined. Unlike models based on Marticorena
and Bergametti [1995] or Shao [2011], threshold values are based on surface and wind tunnel flux measurements of saltation
[Gillette, 1988]. The drag partition in the FENSGHA scheme is described by the MacKinnon et al. [2004] parametrization
using the model surface roughness (z0) or derived from the surface roughness estimates using the Advanced Scatterometer
(ASCAT) as described by Prigent et al. [2012]. The Fécan et al [1998] soil moisture correction is used to adjust the dry
threshold friction velocity. Once the total windblown dust emission flux is computed the total flux is distributed into the
modeled dust bins using the Kok [2011] distribution.
A new sediment supply map, the Baker-Schepanski Map, which was developed from the ideas of Chappell and Webb [2017]
is currently used within the GEFS-Aerosol FENGSHA implementation. Chappell and Webb created an approach similar to
that of the Raupach [1992] model for lateral cover but instead uses a top-down view to describe the area of the turbulent wake
using an analogous shadow instead of a two-dimensional view. The shadow approach is sensitive to the configuration of the
roughness elements meaning that it is sensitive to the placement of the roughness elements in relation to each other. The BSM
describes the probability of momentum mixing directly to the soil surface through the canopy. For the application into GEFS-
Aerosols a monthly 3-year climatology of the BSM was created.
**2.1.3 GEFS-Aerosols Coupled Architecture**
The aerosol component of GEFS-Aerosols couples directly to the FV3-based atmospheric component via the NUOPC Layer
[Theurich et al., 2016], which is the foundation of NOAA's modeling framework (Fig. 1). The aerosol component imports
meteorological fields from the atmospheric model and exchanges aerosol mixing ratios at each coupling time step via standard
NUOPC connectors. Each aerosol species is simulated as a prognostic atmospheric tracer, which is advected by the FV3
dynamical core and undergoes convective mixing and PBL diffusion within the atmospheric physics. All aerosol composition





and emission-related processes are computed in GEFS-Aerosols after the atmospheric physics has been advanced. Tracer mixing ratios are then updated and exported back to the atmospheric model.

Bundling all aerosol composition processes in a single model component led to the implementation of a sequential coupling scheme with the atmospheric component. At each coupling time step, the atmospheric dynamical core and physics processes (including radiation) are computed first. The aerosol component is then executed to perform all air composition processes and transfer the updated tracers back to the atmospheric component. Finally control returns to the atmospheric model, which updates the atmospheric state with new meteorology and aerosols concentrations. To minimize overhead associated with data exchange between model components, GEFS-Aerosols is run on the atmospheric grid, which is imported from the atmospheric component through NUOPC. Additionally, the coupling run sequence assigns to the aerosol component the identical set of Persistent Execution Threads (PETs) used by the atmospheric model's forecast component. This allows the model to leverage NUOPC's ability to access coupling fields by memory reference, minimizing the memory footprint for the coupled system.

The GEFS is a weather forecast modeling system made up of 31 separate forecasts, or ensemble members. The NCEP started the GEFS to address the nature of uncertainty in weather observations that are used to initialize weather forecast models and uncertainties in model representation of atmospheric dynamics and physics. The aerosol component coupled with FV3GFS v15 has been merged into the GEFS, as a single ensemble member named GEFS-Aerosols, for real-time and retrospective forecast that preceded operational implementation, which occurred in September 2020.

The sequence of steps involved in moving from the beginning to the end of a forecast process is controlled by the workflow. In a retrospective or real time forecast, the chemical tracers are cycled from the output of a previous forecast as the initial condition. The workflow shown in Figure 2(b) describes the steps including both pre-processing and post-processing for the GEFS-Aerosols real-time or retrospective forecast. This initial implementation of GEFS-Aerosols does not include assimilation of aerosol observations, so the chemical tracers in the restart files are used as the chemical initial condition for the next forecast. The AOD is calculated in the post-processing part of the workflow, using a look-up table (LUT) of aerosol optical properties from NASA, which has been implemented in the Unified Post Processor (UPP, https://dtcenter.org/community-code/unified-post-processor-upp). It should be noted that the LUT reflects the impacts of a larger number of aerosol species in the atmosphere than the simple GOCART module treats. Some adjustments with scaling factors based on observation have been applied to compensate for the absence of nitrate, ammonia and secondary organic aerosol (SOA) in GOCART.

## 2.2 Observation, reanalysis data and other model data

The real-time forecast experiments were evaluated using the following ensemble analysis, reanalysis data, satellite and in situ observational data, aircraft measurements, and model predictions:

1) International Centers for Aerosol Prediction - Multi-Model Ensemble (ICAP-MME) provides daily 6-hourly forecasts of total and dust AOD globally out to 120 h at 1 degree horizontal resolution [Reid et al., 2011; Sessions et al., 2015;



Xian et al., 2019]. Total AOD in ICAP-MME is provided by the four core multispecies models: the European Centre Medium Range Weather Forecasts Copernicus Atmosphere Monitoring Service (ECMWF-CAMS), the Japan Meteorological Agency Model of Aerosol species in the Global Atmosphere (JMA-MASINGAR), the NASA Goddard Earth Observing System Version 5 (NASA-GEOS-5) and the Naval Research Lab Navy Aerosol Analysis and Prediction System (NRL-NAAPS) modeling systems. Dust-only AOD is provided by the aforementioned four models, plus the Barcelona Supercomputer Center Chemical Transport Model (NMMB/BSC-CTM), the United Kingdom Met Office Unified Model (UKMO-UM) and NGACv2. All four of the multispecies models incorporate aerosol data assimilation (DA) and satellite-based smoke emissions.

2) Total AOD instantaneous reanalysis dataset from the second Modern-Era Retrospective analysis for Research and Application [MERRA-2, Gelaro et al., 2017]. The MERRA-2 reanalysis provides various AOD forecasts at 0.625 x 0.5 degree horizontal resolution and at 72 vertical levels.

3) The Aerosol Robotic Network (AERONET), which is a global ground-based network of automated sun-photometer measurements, provides AOT, surface solar flux and other radiometric products [Holben et al., 1998]. It is a well-established network of over 700 stations globally and its data are widely used for aerosol-related studies [Zhao et al., 2002]. AERONET employs the CIMEL sun–sky spectral radiometer, which measures direct sun radiances at eight spectral channels centered at 340, 380, 440, 500, 675, 870, 940 and 1020 nm. AOT uncertainties in the direct sun measurements are within ±0.01 for longer wavelengths (longer than 440 nm) and ±0.02 for shorter wavelengths [Eck et al., 1999].

4) MODIS provides near-global coverage of aerosol measurements in space and time. We have used a MODIS Level-3 (daily and monthly at 1 degree horizontal resolution) AOD dataset in this study (https://ladsweb.nascom.nasa.gov/). The dataset belongs to the Collection 6.1 combined land and ocean from the Aqua satellite [Levy et al., 2013]. This latest collection of MODIS data includes AOT data based on refined retrieval algorithms, in particular the expanded Deep Blue algorithm [Hsu et al., 2013; Sayer et al., 2013]. It introduces a merged AOD product, combining retrievals from the Dark Target (DT) and Deep Blue (DB) algorithms to produce a consistent data set covering a multitude of surface types ranging from oceans to bright deserts [Sayer et al., 2014]. In this work the aerosol product Dark_Target_Deep_Blue_Combined_Mean was used for quantitative evaluation of model results.

5) The Visible Infrared Imaging Radiometer Suite (VIIRS) sensor onboard the Suomi National Polar Orbiting (S-NPP) satellite provides sets of aerosol environmental data records (EDRs) based on daily global observations from space [Jackson et al., 2013; Liu et al., 2013]. Beginning in 2012, VIIRS provides AOT at 550 nm at a global 0.25 degree horizontal resolution. Daily gridded Enterprise Processing System (EPS) VIIRS data used are from the NOAA STAR ftp site at ftp://ftp.star.nesdis.noaa.gov/pub/smcd/VIIRS_Aerosol/npp.viirs.aerosol.data/epsaot550/ .

6) The NASA Goddard Earth Observing System (GEOS), Version 5 (GEOS-5) model prediction. The GEOS model consists of a group of model components that can be connected in a flexible manner in order to address questions related to different aspects of Earth Science (https://gmao.gsfc.nasa.gov/GEOS/).





7) The NEMS GFS Aerosol Component Version 2.0 (NGACv2) for global multispecies aerosol forecast developed by NCEP and collaborators was previously used to provide operational global multispecies aerosol forecasts at NCEP [Wang et al., 2018]. The anthropogenic emissions are based on EDGAR V4.1 [Janssens-Maenhout, 2010] and AeroCom Phase II [Diehl et al., 2012]. The smoke emissions are from NESDIS STAR's GBBEPx, blended from the global biomass burning emission product from a constellation of geostationary satellites [GBBEP-Geo, Zhang et al., 2012; Zhang et al., 2014] and GSFC's Quick Fire Emission Data Version 2 from a polar-orbiting sensor [QFED2, Darmenov and da Silva, 2015]. NGACv2 uses the same physics package as the 2015 version of the operational GFS.

8) The Atmospheric Tomography Mission (ATom) studies the impact of human-produced air pollution on greenhouse gases and on chemically reactive gases in the atmosphere [Wofsy et al., 2018]. ATom deploys instrumentation to sample atmospheric composition, profiling the atmosphere in 0.2 to 12 km altitude range. Flights took place in each of 4 seasons over a 22-month period in 2016 through 2018. They originated from the Armstrong Flight Research Center in Palmdale, California, flew north to the western Arctic, south to the South Pacific, east to the Atlantic, north to Greenland, and returned to California across central North America over the Pacific and Atlantic oceans from ~ 80°N to ~ 65°S. ATom establishes a single, contiguous global-scale data set. This comprehensive data set is used to improve the representation of chemically reactive gases and short-lived climate forcers in global models of atmospheric chemistry and climate. The Particle Analysis by Laser Mass Spectrometry (PALMS) instrument samples the composition of single particles in the atmosphere with diameters within ~150 nm - 5 μm range.

The Particle Analysis by Laser Mass Spectrometry (PALMS) instrument samples the composition of single particles in the atmosphere with diameters within ~150 nm - 5 μm range. It measures nearly all components of aerosols from volatiles to refractory elements, including sulfates, nitrates, carbonaceous material, sea salt, and mineral dust [Murphy et al., 2006]. The PALMS instrument was originally constructed for high-altitude sampling [Thomson et al., 2000; Murphy et al., 2014] and has since been improved and converted for other research aircraft. Uncertainty in mass concentration products is driven mainly by particle sampling statistics. Relative 1sigma statistical errors of 10-40% are typical for each 3-min sample at a mass loading of 0.1 ug/m3 [Froyd et al., 2019]. In August 2016, PALMS was sampling on the NASA DC-8 aircraft as part of the ATom program (https://espo.nasa.gov/missions/atom/content/ATom). Aerosol composition determinations using the PALMS instrument during ATom have been described and interpreted previously [Murphy et al., 2018, 2019; Schill et al., 2020; Bourgeois et al., 2020]. The PALMS mass concentrations for various species are derived by normalizing the fractions of particles of each size and type to size distributions measured by optical particle counters [Froyd et al., 2019].

## 3. Background Fields and Emissions

### 3.1. Anthropogenic emissions and background fields



The preprocessor PREP-CHEM-SRC v1.7, a comprehensive tool that prepares emission fields of trace gases and aerosols for
use in atmospheric chemistry transport models, was used to generate the anthropogenic emissions, background fields of OH,
$H_2O_2$, $NO_3$, DMS and dust scheme input of clay and sand at the FV3 grid resolution for GEFS-Aerosols [Freitas et al., 2011].
Two global anthropogenic emission inventories were chosen as input to drive the model, both providing monthly emissions.
One is from the Community Emissions Data System (CEDS), which provides the emissions of BC, OC and $SO_2$ in 2014
[Hoesly et al., 2018]. The CEDS inventory improves upon existing inventories with a more consistent and reproducible
methodology applied to all emission species, updated emission factors, and more recent estimates in 2014.  The data system
relies on existing energy consumption data sets and regional and country-specific inventories to produce trends over recent
decades [Hoesly et al., 2018]. The Hemispheric Transport of Air Pollution (HTAP) version 2 [Janssens-Maenhout et al., 2015]
inventory provides the emissions of BC, OC $SO_2$, $PM_{2.5}$ and $PM_{10}$ in 2010.
Figure 3 shows the comparisons of anthropogenic emissions between CEDS and HTAP for $SO_2$, BC and OC in July. Aside
from the shipping lanes showing up in CEDS, there is generally broader spatial coverage in CEDS. For $SO_2$, the CEDS
emissions are much larger over the eastern US, eastern China and Europe.  Much higher values of BC and OC are seen in
CEDS over Eastern Asia, South Asia and Europe. Similarly, much larger values for BC and OC are seen in the Southern
Hemisphere in CEDS. We performed experiments by comparing model predictions using these two different anthropogenic
emissions datasets with ATom-1 observations (figures not shown here). Slight improvements in $SO_2$ correlations and bias are
seen and the sulfate, OC and BC biases improve over the Atlantic Ocean when using the CEDS emissions in comparison with
the HTAP dataset.
The GOCART model background fields have been replaced by the newer version of 2015 from the NASA GEOS/GMI model.
We validated the GOCART background fields of OH and $H_2O_2$ against the ATom-1 observations. Even though these
background fields are model-derived climatologies, they both compare very well with the ATom-1 measurements. The newer
NASA GEOS/GMI fields show improvement in the model-measurement biases for both OH and $H_2O_2$.
**3.2 Biomass burning emission**
There are two biomass burning emission options in the model. One is using the GBBEPx V3 emission with Fire Radiative
Power (FRP). The GBBEPx V3 system produces daily global biomass burning emissions of $PM_{2.5}$, BC, CO, $CO_2$, OC, and
$SO_2$) by blending fire observations from MODIS Quick Fire Emission Dataset (QFED), VIIRS (NPP and JPSS-1) fire
emissions, and Global Biomass Burning Emission Product from Geostationary satellites (GBBEP-Geo). GBBEP-Geo also
produces hourly emissions from geostationary satellites, at individual fire pixels. In the results shown here, GBBEPx daily
biomass burning emissions on the FV3 C384 global grid are used for GEFS-Aerosols. The details of the GBBEPx V3 algorithm
can be found in  https://www.ospo.noaa.gov/Products/land/gbbepx/docs/GBBEPx_ATBD.pdf.
The other biomass burning emission product is estimated by the Brazilian Biomass Burning Emissions Model (3BEM) model
[Longo et al. 2010; Grell et al., 2011], which is based on near real-time remote sensing fire products of MODIS to determine





fire emissions and is replaced by the Wildfire Automated Biomass Algorithm (WF_ABBA) processing system over the
Americas.
A one-dimension (1-D) time-dependent cloud module from High-Resolution Rapid Refresh (HRRR)-Smoke model has been
implemented into GEFS-Aerosols to calculate injection heights and emission rates online [Freitas et al., 2007]. The new
scheme in HRRR-Smoke is a modified version of the 1D plume rise scheme used in WRF-Chem [Freitas et al., 2007]. The
new plume rise scheme is using the FRP data instead of the look-up table to estimate the fire heat fluxes [Ahmadov et al,
2017]. The 1-D cloud module is able to be applied with these two different fire emissions datasets to account for plume-rise
that distributes the fire emissions vertically and better simulate the fire events and pollution transport of smoke plumes.
**3.3 Impacts of Different Fire Emission Inventories and Plume-rise and fire events evaluation**
Real-time estimates of biomass burning emissions for predictions are based on real-time satellite observations. Challenges for
these emissions involve detection of fires, the strength and composition of the emissions, altitude of the plume rise, temporal
distribution of the emissions and the uncertainty in persistence or change of emissions during the forecast period. From the
comparisons of OC fire emission between GBBEPx V3 and MODIS plus WF_ABBA on June 30 2019 (see Figure 4), we can
see that GBBEPx V3 fires emissions exhibit more fires locations/counts, with wider spatial coverage and more fire emissions
over western North America, Eastern Europe, South America, Asia, and southern Africa, than that of MODIS plus WF_ABBA
emissions. Generally, the GBBEPx V3 emissions are much larger over Eastern Europe, Alaska, South Asia and South America
than those from MODIS. In contrast, MODIS plus WF_ABBA emissions are much stronger over Southern Africa even though
they have a smaller spatial extent.
To quantify the impact of different fire emissions and plume-rise on the fire aerosol forecast, three sensitivity experiments
were performed in real-time starting on June 1, 2019, with the same aerosol initial conditions:
**Exp.1)** MODIS plus WF_ABBA fire emissions combined with the plume-rise module based on prescribed parameters;
**Exp.2)** GBBEPx V3 fire emissions emitted at the surface without the plume-rise module;
**Exp.3)** GBBEPx V3 fire emissions combined with the plume-rise module and real-time FRP data.
Figure 5 shows the Total AOD of these three experiments calculated by GEFS-Aerosols UPP and compared with ICAP and
NGACv2 on June 30, 2019.  Generally, the spatial patterns in AOD from the three experiments are quite similar with maxima
over southern Africa and South Asia. Comparing the fires OC emission distributions (see Figure 4), the OC fire emissions of
GBBEPx V3 are smaller over southern Africa than that of MODIS plus WF_ABBA, producing a much lower OC AOD in
Exp.2 than that of Exp.1. Exp.3 showed similar spatial distributions to that of Exp.1 although with a larger spatial extent of
high AOD. The larger extent is due to the inclusion of the FRP-based plume rise algorithm in Exp.3 lifting the emissions up
to higher altitudes where long-range transport has more of an effect.
Comparisons of ICAP and the three GEFS-Aerosols experiments are shown in Figure 5 (the former minus latter). AOD is
overpredicted by GEFS-Aerosols relative to ICAP over most of the tropical areas (e.g. southern Africa and south Asia) due to





biomass burning sources but underpredicted over eastern Europe and northwest America by GEFS-
Aerosols Exp.1. GEFS-Aerosols performance relative to ICAP is improved when using the GBBEPx V3 emissions, with the discrepancies decreasing
compared to using the MODIS emissions, especially the underprediction over eastern Asia. Overall, the GEFS-Aerosols
underprediction is smaller in Exp.3. High AOD from a large fire event in eastern Asia and transported over Alaska is observed
in the ICAP forecast. This event was not captured well in Exp.1, but is captured in Exp.2 and Exp.3 when using GBBEPx V3
emissions, with the largest AOD values occurring in Exp.3 when a plume-rise scheme using real-time FRP is employed.
The right column of Figure 5 shows comparisons between the total AOD from NGACv2 and the ICAP analysis (ICAP minus
NGACv2). The NGACv2 model is also using the GBBEPx V3 without applying any plume-rise module similar to Exp.2.
Discrepancies between the NGACv2 model results and the ICAP analysis are more substantial than between ICAP and GEFS-
Aerosols, with wide NGACv2 underprediction over east and south Asia, the Southern Hemisphere, eastern US and
northwestern America.
Due to its generally the best performance, the wildfire emissions setup in Exp.3 was adopted for retrospective and operational
forecasts in GEFS-Aerosols. Section 5 will show further comparisons between model configurations with MODIS and
GBBEPx V3 emissions that were evaluated using the ATom-1 measurements.
To further validate model performance when using the GBBEPx V3 fire emissions with a plume-rise module based on real-
time FRP data (Exp.3), we compare real-time GEFS-Aerosols AOD with other reanalysis data, satellite observations and the
NGACv2 model for the fire event in August 2019. Smoke from large fires burning in the Amazon rainforest, primarily in
Brazil, Bolivia, Paraguay, and Peru, stretched over northern South America in mid-August. Figure 6 shows the total AOD
forecast on 25th August compared against the NGACv2 model, MERRA-2 reanalysis data and satellite observations of VIIRS
and MODIS. For both satellites, daily gridded AOD is used to compare against the model forecast at 18z. The GEFS-Aerosols
AOD is able to reproduce the enhanced AOD due to several fire events over South America near the border of Bolivia,
Paraguay, and Brazil, which were also observed by the VIIRS and MODIS satellite instruments and captured by the MERRA2
analysis.  Although there are a lot of missing data downwind from the fires in the satellite observations of VIIRS and MODIS,
especially over the south Pacific, GEFS-Aerosols and MERRA-2 results are consistent in showing the transport of fire plumes
into the tropical Pacific and southern Atlantic. In contrast, the NGACv2 model does not capture these fire events, and exhibits
only a very slight AOD enhancement. NGACv2 AOD is more than 80% smaller than the observations over the fire source
region and produces little or no transported smoke over the surrounding areas.
Beyond the fires burning in South America, an even greater number of blazes on the African continent are observed by the
satellite images at almost the same time in August 2019. Angola experienced almost three times more fires than Brazil in the
middle of August 2019. There were around 6,000 fires in Angola, more than 3,000 in Congo and just over 2,000 in Brazil,
according to NASA satellite imagery (https://earthobservatory.nasa.gov/images/145421/building-a-long-term-record-of-fire).
One of the main large-scale aerosol features of Sub-Saharan Africa is the June-to-September biomass burning season in Angola,
Congo, and Zambia [Bauer et al., 2019]. Overall, the GEFS-Aerosols model reasonably simulates the major burning event on
August 25th, 2019 over southern Africa (Figure 6), but overestimates the central African plume when compared with the





MERRA2 analysis. The satellite AOD retrievals of VIIRS and MODIS off the coast of central Africa are challenging due to
screening by the stable stratiform cloud deck over the ocean that occurs during the fire season, creating less reliable coverage
from these observational data. Nevertheless, we can still see consistent AOD enhancements over the fire source regions and
surrounding areas for both GEFS-Aerosols and the observations. NGACv2, however, is quite different from the satellite
observations and MERRA2 analysis, underestimating the AOD more than 50-90% percent over the southern Africa fire source
region and showing little obvious enhancement.
Future work will explore the use of diurnal fire profiles based on historic GOES fires products applied to estimate biomass
burning emissions to enhance forecast behavior. Additionally, a parameterization based on fire weather index (FWI) to
estimate biomass burning emissions on longer temporal scales may help to improve and extend the forecast of fire impacts.
**4. Real-time forecast Evaluation**
**4.1 Evaluation of global AOD**
A real-time forecast was performed starting on July 1, 2019 and continuing to the present day using the GBBEPx V3 fire
emissions with the plume-rise module based on real-time FRP data. Figure 7 shows the Day 1 AOD prediction of GEFS-
Aerosols and NGACv2 compared with the MERRA-2 reanalysis and MODIS observations averaged from July to November
2019. The GEFS-Aerosols prediction is able to capture the prominent temporal and geographical features of AOD as
represented by the MERRA-2 reanalysis data and MODIS satellite observations, such as the dust plumes over North Africa
and the Arabian Peninsula, biomass burning plumes in southern Africa, south America, northwestern America and eastern
Europe, polluted air over eastern and southern Asia, and high-latitude sea-salt bands over the southern hemisphere. The high
AOD over southern Africa and northern India is more comparable to the MODIS observation. As pointed out by Bhattacharjee
et al. [2018], the NGACv2 predictions exhibit widespread underestimates over most of these high AOD regions, such as eastern
Asia, and fire source regions of southern Africa, eastern Europe, southeastern Asia.
Figure 8 indicates the Day 1 AOD forecast biases of GEFS-Aerosols and NGACv2 with respect to GEOS-5 analysis between
7/5/2019 and 11/30/19 for dust, OC and sulfate. The predicted dust AOD in GEFS-Aerosols is quite comparable to that of
GEOS-5 results, with only small negative biases of ~0.08 over Asia and the downwind areas of African dust source regions of
Atlantic and south Asia (Figure 8a). GEFS-Aerosols has some small positive biases relative to GEOS-5 of ~0.1 over part of
North Africa and Australia (in red). In contrast, dust AOD in NGACv2 (Figure 8b) shows large over predictions of GEOS-5
over Africa with maximum value ~ 0.45 and about 0.02-0.05 over large areas of Asia and North Pacific and North America.
Wang et al. [2018] also showed that the predicted dust AOD in NGACv2 over northwestern Africa is much larger than GEFS-
Aerosols, MERRA-2 and MODIS observations.
OC is a major component emitted from wildfires, and OC AOD is a good indicator of the performance of fire impacts. GEFS-
Aerosols OC AOD shows smaller biases compared to the GEOS-5 analysis than those of NGACv2 (Figure 8c and d). Positive
biases in GEFS-Aerosols OC AOD of less than 0.2 occur mainly over southern Africa, eastern Asia, south Asia and the Middle


East. The GEFS-Aerosols overprediction of OC AOD compared with GEOS-5 over eastern China may be associated with the
overestimate of anthropogenic emissions, since this is not a major fire source region. GEFS-Aerosols shows small negative
biases, of less than 0.1, over South America and middle and eastern Europe. Overall, the biases of OC AOD in NGACv2
relative to GEOS-5 are dominated by under prediction globally with the largest biases of more than 0.3 over major fire source
regions of southern Africa, the Amazon region of South America, Southeastern Asia and Sibera (Figure 8d).
For sulfate AOD, the GEFS-Aerosols forecast over predicts GEOS-5 by ~0.08 over eastern Africa, the Middle East, and south
eastern China, where $SO_2$ anthropogenic emissions are dominant. Small GEFS-Aerosols under predictions, less than 0.1 AOD,
are seen over broad areas of the Northern Hemisphere, such as eastern North America and its downwind areas over the north
Atlantic and western Europe, as well as eastern Asia and its downwind areas (Figure 8e). As in the case of OC AOD, the global
sulfate AOD in GEOS-5 is under predicted significantly by NGACv2 (Figure 8f). The areas with the largest NGACv2 vs.
GEOS-5 sulfate bias are mainly over the major anthropogenic source regions, such as India and eastern China, where the
underestimates exceed 0.18, and in the eastern US and western Europe, where they exceed 0.1.
The summary comparison of the GEFS-Aerosols and NGACv2 Day 1 total AOD prediction biases with respect to MERRA-2
analysis between 7/5/19 and 11/30/19 is shown in Figure 9. Generally, the GEFS-Aerosols model is able to reproduce the total
AOD very well, much better than NGACv2 (see Figure 9a and b). The GEFS-Aerosols over predictions over eastern China
and Southern Hemisphere (~0.2-0.3) are mainly due to anthropogenic OC and $SO_2$ for the former and fire emissions of OC for
the latter, respectively. Both GEFS-Aerosols and NGACv2 total AOD have  small negative biases (~0.1) relative to MERRA-
2 over the northwestern China dust source region. Negative biases of GEFS-Aerosols vs. MERRA-2 in South America may
be caused by inadequate fire emissions and in Europe that may be related to anthropogenic $SO_2$ emissions. The spatial locations
of biases in GEFS-Aerosols with respect to MERRA-2 analysis total AOD (Figure 9a) are similar to the comparisons with the
individual aerosol species from GEOS-5 discussed above (Figure 8a, c and e). The NGACv2 total AOD is biased low relative
to MERRA-2 almost globally except for the over prediction over North Africa due to dust (Figure 9b). The largest NGACv2
total AOD biases are mainly caused by the under predictions of fires over the fire source regions of South America, southern
Africa, Southeastern Asia and middle and eastern Europe and the anthropogenic source regions over eastern China, India and
eastern North America, with maximum total AOD biases reaching more than 0.5.
In addition to comparing with the GEOS-5 and MERRA-2 data, we also evaluated the GEFS-Aerosols model performance
with the daily AERONET data globally. The locations of the 60 selected AERONET sites where these comparisons were made
are listed in Table 1. The GEFS-Aerosols, NGACv2 and ICAP predictions are sampled at the same locations as the AERONET
sites for these comparisons. The top 2 panels in Figure 10 show correlation coefficients between daily total AOD observed by
AERONET and the day 1 forecast of model AOD from GEFS-Aerosols and NGACv2 for the period between 7/5/2019 and
11/30/19. The correlation coefficients range from 0.5 to 0.9 for GEFS-Aerosols at most sites, except for several sites in South
America, Africa and eastern Asia near fire source regions, which are slightly lower than those of the ICAP. In contrast, the
correlation coefficients of daily total AOD between the NGACv2 and AERONET observations are lower than 0.5 globally,
even ranging from 0.1 to 0.3 at most sites. A more quantitative display of correlation coefficients for a selection of 30





AERONET sites for GEFS-Aerosols and NGACv2 is presented in the bottom panel of Figure 10. This comparison strongly
indicates the improved performance of total AOD daily variation in GEFS-Aerosols prediction when compared with NGACv2.
There are 20 sites (about 30% of total sites) displaying highly correlated total AOD for the AERONET data and GEFS-
Aerosols, with the correlation coefficients exceeding 0.7. In contrast, there is only 1 site with a correlation coefficient larger
than 0.7 for NGACv2 model vs. AERONET, and 19 sites have correlation coefficients that are less than 0.2 for AERONET
and NGACv2.

**4.2 Evaluation of AOD associated with fire events**

Figure 11 indicates the total AOD time series of AERONET observations compared against ICAP, NGACv2 and GEFS-
Aerosols model predictions at the four AERONET sites near the fire source regions of South America during the period of
7/1/19-11/30/19. At the Alta Floresta site, which is in the middle of Amazon fire source region, the daily AOD variations of
both the ICAP and GEFS-Aerosols day 1 predictions are quite consistent with that of the AERONET data, especially as they
are able to reproduce two peaks in AOD enhancements in late August and late September caused by fire plumes (Figure 11a).
NGACv2 results under predict AERONET observations almost throughout the whole period with a significantly larger bias
than GEFS-Aerosols or ICAP, and the two August-September peaks in total AOD enhancements are essentially missed in the
NGACv2 prediction.
The Itajuba site is located southeast of the Alta Floresta site and in the downwind areas of the Amazon fire source region.
While the total AOD time series of GEFS-Aerosols prediction matches closely those of ICAP and AERONET during most of
the time period, there are some discrepancies from the end of August to middle September when the GEFS-Aerosols
underpredicts the high AOD episode (Figure 11b). However, GEFS-Aerosols is able to predict the other two AOD
enhancements in mid-October and early November, performing better than ICAP. The NGACv2 prediction also generally
underestimates the observations at this site too (Figure 11b). The NGACv2 results are closer to the ICAP, GEFS-Aerosols and
AERONET before August, and NGACv2 shows a slight increase of total AOD in early September, but the NGACv2 AOD
magnitude is much lower than the AERONET by about a factor of 5-7 from the middle of August onward.
Located in the southern part of the Amazon fire region, the site of Santa Cruz Utepsa is south of the AltaFloresta site. The
GEFS-Aerosols model shows very good performance at this site in predicting the total AOD through the 5 months from July
to November (Figure 11c). The model not only reproduces the total AOD temporal variation of the AERONET results, but
also captures several fluctuations of high AOD in August and September caused by Amazon fire events. Again, some of
fluctuations in total AOD were captured by the NGACv2 prediction, but the modeled AOD magnitudes are 2-4 times lower
than the observations.
The last site of RioBranco is also located in the Amazon fire source region but to the west of the AltaFloresta site. There are
some missing data at this site for the AERONET total AOD from middle July to middle September (Figure 11d). During this
period, the GEFS-Aerosols prediction is comparable to that of ICAP and slightly lower than ICAP by about 5-10%. Both ICAP





and GEFS-Aerosols total AOD match the AERONET variations well when the AERONET data are available again from mid-September. Several peaks of total AOD are also captured by GEFS-Aerosols in middle September and early November. The NGACv2 prediction shows enhanced total AOD in middle August, with low biases by more than 2-3 times compared to ICAP and GEFS-Aerosols. For other enhancements of total AOD after October, the NGACv2 results completely miss the fire events and do not show any fluctuations.

We also evaluate the total AOD time series of AERONET against ICAP, NGACv2 and GEFS-Aerosols for fire regions of central and southern Africa. The comparisons at eight AERONET sites from July to September are shown in Figure 12. Generally, the GEFS-Aerosols predictions are able to capture the daily total AOD variation measured by AERONET somewhat better than that of ICAP at most of the sites near the fire source regions, except at the station Bamenda located north of the major African fire source region. The NGACv2 total AOD forecast shows under prediction at most of the AERONET sites in this region. The GEFS-Aerosols prediction overestimates the total AOD most of time throughout the three months at the Bamenda site. Meanwhile, NGACv2 and ICAP predictions are not consistent with AERONET either, especially for several observed high peaks which are not reproduced by any of the model results. At the remote site of Ascension Island located west of the African fire source region, the GEFS-Aerosols model is able to capture the AOD enhancements in the middle of August, and shows the best performance of the three different models. For other sites that are located in the fire source region, such as Monguinn, Misamptu, Maun Tower, and Lubango, the prediction of the GEFS-Aerosols model shows very good performance and matches the observed temporal variation of total AOD very well. One peak in early August at the Monguinn site, one peak in middle September at the Misampfu site, two peaks in early August and early September at Maun Tower site and one enhancement in August at Lubango are all predicted very well by the GEFS-Aerosols model. The ICAP forecasts still have lower biases against the AERONET total AOD for predicting these peaks, but none of these peaks are captured by NGACv2. While GEFS-Aerosols shows good performance compared to AERONET, there are slight over predictions in mid-July and late August for Gabon and early August for Lubango.

**4.3 Evaluation of AOD associated with dust events**

Thirteen AERONET sites inside the major dust source regions of western North Africa, Asia and the Middle East and surrounding areas have available data from July to November 2019. The total AOD time series of GEFS-Aerosols, ICAP, and NGACv2 at 6 of these sites are shown in Figure 13. Overall, the GEFS-Aerosols model is able to closely predict the observed total AOD variation, especially at the sites of Banizoumbu, Tenerife, Saada, Ben Salem, Granada and Sede Boker, with much better performance than those of NGACv2. In addition to NGACv2's overprediction at the sites of Ben Salem and Granada, it does not accurately capture observed temporal variations of total AOD at these sites.

We compare the daily AERONET total AOD with the 1-day forecasts of total AOD from GEFS-Aerosols and NGACv2 at the AERONET sites of Cape Verde, Tamanrassett and Tenerife located in the dust source region over North Africa in Figure 14. The slope of the linear regression of AERONET total AOD vs. GEFS-Aerosols is quite different from that of NGACv2 at the





site of Tamanrassett, which is located in southern Algeria and in the middle of the Saharan dust source region. The GEFS-Aerosols linear regression slope is much closer to 1 than that of NGACv2, and the $R^2$ in the NGACv2 model is lower by a factor of 4 than that of the GEFS-Aerosols model at this site. At the other two sites of Cape Verde and Tenerife, which are in the downwind area west of the African dust source region, the slopes of the linear regressions for GEFS-Aerosols are also much closer to 1 than those of the NGACv2 model. The NGACv2 model, as evidenced by the $R^2$ values, is more poorly correlated with AERONET than the GEFS-Aerosols prediction, which better captures the dust transport in the downwind areas west of the African dust source region than the NGACv2 model.

### 4.4 Evaluation of major regional averages

Figure 15 shows day 1 predictions of total AOD time series by GEFS-Aerosols and NGACv2 compared against the MERRA-2 reanalysis averaged over 9 major global regions from August 2019 to March 2020. The comparison clearly shows the consistency between GEFS-Aerosols and the MERRA-2 reanalysis over most of these 9 regions, especially North Africa, the North Atlantic, Southern Africa, and the South Atlantic, with only minor discrepancies during these 8 months. The total AOD is dominated by dust in North Africa and fire emissions in Southern Africa. The aerosols emitted from dust and fire regions and their long-range transport play important roles in impacting the total AOD over the North and South Atlantic Oceans. The good agreement with MERRA-2 shows that GEFS-Aerosols captures the emissions and transport of dust and fire emissions in these regions.

Total AOD variation in South America is mainly related to biomass burning emissions. GEFS-Aerosols has some slight low biases relative to MERRA-2 from middle September to early October 2019 that are associated with the Amazon fire event. GEFS-Aerosols under predicts MERRA-2 in this region from mid-November 2019 to March 2020, outside the main biomass burning season, which suggests that the GEFS-Aerosol AOD low biases in this region are mostly associated with sources other than fires.

The European region has the largest differences between GEFS-Aerosols and MERRA-2 reanalysis total AOD among the 9 regions. Although their temporal variations are similar, GEFS-Aerosols under predicts the MERRA-2 total AOD throughout the whole period by a factor of 0.5. The large absolute low biases from August to early October 2019 and March 2020 in Europe which are associated with GEFS-Aerosols underestimates in dust emissions.

From August to early December 2019, the GEFS-Aerosols total AOD looks quite consistent with the MERRA-2 reanalysis on average across East Asia. GEFS-Aerosols high biases starting in middle December 2019 and increasing from January to March 2020 may be associated with the lockdown in China during the Coronavirus disease 2019 (COVID-19) pandemic. Anthropogenic emissions of $NO_2$, $SO_2$, VOC, and primary $PM_{2.5}$ over the North China Plain during this period were reduced by 51%, 28%, 67% and 63%, respectively, compared to the previous year, resulting in lower surface aerosol and ozone levels and improvements to air quality [Shi and Brasseur, 2020; Wang and Su, 2020; Xing et al., 2020]. Since the anthropogenic



emissions used in GEFS-Aerosols are based on the CEDS 2014 inventory, they definitely overestimate the anthropogenic aerosol emissions during the 2019-2020 lockdown periods.

Both the Eastern and Western US regions exhibit GEFS-Aerosols low biases of about 5-30%, with the largest differences in Eastern US occurring in August 2019. However, the total AOD temporal variations in the GEFS-Aerosols prediction and the MERRA-2 reanalysis are quite consistent over Eastern and Western US. The minor under predictions by GEFS-Aerosols may be a combined effect of both errors in the anthropogenic emissions and uncertainties in aerosol wet removal processes.

In comparison, the NGACv2 predictions show significant under prediction of MERRA-2 total AOD for almost all of these 9 regions throughout this 8-month period. The one exception is North Africa, where the NGACv2 results are close to the MERRA-2 reanalysis, with over prediction in August 2019 and low biases from December 2019 to March 2020. In addition to its general under prediction of MERRA-2 total AOD, NGACv2 is not able to capture the temporal variations of total AOD in some regions, such as the enhanced AOD due to fire emissions in Southern Africa, South Atlantic and South America. Though NGACv2 shows similar temporal variations to MERRA-2 total AOD in Europe, East Asia, and the US, the magnitudes of NGACv2 predictions are too low by a factor of 1 to 3. This analysis is consistent with a 1-year evaluation of GEFS-Aerosols AOD that shows improvements over NGACv2 [Bhattacharjee et al., 2021]

## 5. ATom-1 retrospective forecast evaluation

Retrospective simulations of GEFS-Aerosols and NGACv2 were performed for the summer of 2016 and evaluated using aircraft measurements from the first deployment of the Atmospheric Tomography Mission (ATom-1) in July and August 2021. During ATom-1, plumes from dust storms and large biomass burning events and low-level sea salt aerosols were observed over the southern and central Atlantic, and anthropogenic pollution was observed over the United States on the last flight from Minnesota to Southern California.

In this section, we evaluate the 24-hour forecast skill of GEFS-Aerosols and NGACv2 by comparing with ATom-1 observations. The GEFS-Aerosols and NGACv2 model results are sampled at the same latitude, longitude and altitude as the ATom-1 measurements. The model output is hourly on the FV3 native grid. The ATom-1 measurements collected on a 1-second time base were compared to the nearest hour's model forecast. Model data is interpolated vertically (according to log-Z AGL), but sampled within the nearest horizontal grid as the observations (with no horizontal interpolation). Thus, the inherent differences between temporal (differences of up to 0.5 hour) and spatial scales of the observations (~200 m resolution) and model results (25-100 km resolution) must be kept in mind with the model-measurement comparisons. To quantify the impact of model resolution, we performed simulations at FV3 horizontal resolutions of C96 (~100 km) and C384 (~ 25km) using the GBBEPx V3 fires emissions with a plume-rise module based on real-time FRP data.

### 5.1 Global flight track column sum comparisons



Figure 16 shows the tropospheric column sums of OC along the flight tracks of the NASA DC-8 for the ATom-1 observations
and two GEFS-Aerosols model experiments. The OC column sums in the two experiments using GBBEPx V3 fire emissions
at either C96 or C384 resolution (Fig. 16 (b) and (c)) are quite consistent and comparable to the observations. Though the
patterns of these two experiments are quite similar, the modeled OC column sums at C384 resolution are somewhat smaller
than those at the C96 resolution over the north Atlantic, Greenland, and southeastern Canada.
Results of the model-measurement comparisons for dust are shown in Figure 17. Both C384 and C96 resolution GEFS-
Aerosols simulations show good agreement with ATom-1 observations over tropical north Atlantic and downwind of the
western Africa dust source region.  However, the model at both resolutions overestimates the dust columns over tropical south
Atlantic, Greenland, southeastern Canada, while underestimating dust over the US, Alaska and broad areas of the Pacific
Ocean. The GEFS-Aerosols model at both resolutions shows a clear enhancement of the dust event sampled on 8/17/16 east
of the African coastline near $22^{\circ}$N, though the model column maxima tend to be more than a factor of 5 lower that the
observations.
Table 2 gives median bias and correlation statistics for column sums of all GEFS-Aerosols model cases as well as the NGAC
dust forecasts for ~130 profiles illustrated in Figures 16 and 17. Correlations (r – Pearson correlation coefficients) are typically
above 0.7 for all species except dust. The OC differences noted above for GBBEPx are apparent in the bias statistics of OC (a
factor of 2.5) and BC (a factor of 50%), although R-correlations are not significantly affected. Differences in the fire inventories
also affect sulfate biases slightly (12%). For all species except for dust, decreases in median model/observed ratios are seen
for the C384 case relative to the C96 case with the same GBBEPx fires. The only difference between these cases is model
resolution, which will have different vertical transport and precipitation scavenging characteristics going from coarser to finer
scale, with a net effect of decreasing the total column amount.
Dust, on the other hand, shows a slight increase in column amount going from C96 to C384. Dust sources depend critically on
surface wind speed, have very little overlap with the anthropogenic and biomass-burning sources of the other species, and are
associated with areas of low precipitation, all which may contribute to the different response of dust to model resolution.
Correlations of dust are also much lower than for other species, and there is a very obvious difference between GEFS-Aerosols
and NGAC model forecast statistics as discussed further below. We note that sea-salt columns are not calculated or compared
to ATom observations, due to the large amount of observations below the detection limit, especially above 2 km altitude.
**5.2 Vertical profile statistics comparisons**
ATom-1 flight tracks are separated into 2 sections and labeled as the "Pacific" side for July 29 to August 8 2016 flights and
the "Atlantic" side for August 15 to August 23 2016 flights. For this analysis the 1-second model and observed data are binned
into 10 equally spaced vertical intervals (~1km) covering the vertical extent of the ATom-1 profiles. Figure 18 shows median,
bias and correlation statistics of OC, BC and sulfate for the 2 geographic regions and the two model cases. For OC over the
Pacific, the median values of the two experiments are lower than that of the observation by more than 50%. Their correlations





with observation are quite similar above ~3 km height, but the C96 model run with GBBEPx v3 fire emissions is much higher below ~3 km height. The vertical profile of bias also suggests that the OC concentrations are underpredicted over Pacific for all of the three experiments. Statistics for the Atlantic flight tracks of ATom-1 show similar trends and behavior. The median values of model prediction OC are quite comparable to observations in both C384 and C96 cases, which show very consistent vertical variation similar to the observations. The correlations with observations improve below 4 km height compared to those of the Pacific with the maximum more than 0.80 in the C96 run. While the correlations decrease significantly at 3-6 km height, it increases almost 50% above 6km height. The C96 experiment is biased too high while the C384 experiment is biased too low above ~3km. Both experiments are biased too low below ~3km. The BC vertical profile statistics are quite different to OC with the model forecasts much larger than observations as one goes higher in altitude over both the Pacific and Atlantic section. Correlation coefficients with observations are about 0.6-0.8 from near surface to ~ 6km over Pacific and about 0.7-0.9 below ~4km over Atlantic. The correlations of both OC and BC decrease significantly above ~4km over the Atlantic. For sulfate, the median biases in the two experiments are all biased too low from near surface to ~ 8 km over the Pacific and their correlation coefficients with observations are about 0.4-0.6 below 6 km. For the Atlantic, median values are quite comparable to the observation below ~7 km and their correlations are about 0.3~0.9 and consistently decreasing with altitude. Generally, the C96 experiment performs better than the C384 experiment. By contrast, the Pacific comparisons for sulfate show a significant underprediction (60-70%) from the 0.5 to 7.0 km altitude for both model cases, which suggests a significant underprediction of oceanic gas-phase sulfur sources, such as DMS.

Vertically resolved statistics of naturally occurring dust and sea salt are shown in Figure 19. For dust over the Pacific, median values of GEFS-Aerosols are too low while the NGACv2 results are too high compared with the observations, and all of the correlations are almost less than 0.5. The performance of GEFS-Aerosols improves over the Atlantic with median values comparable to observations above ~4km and the correlation coefficients increasing to 0.5~0.8 below ~5km, not too much different between the C384 and C96 experiments, but still showing a significant high bias for the NGACv2 model over Atlantic. For sea salt the median biases in the C96 and C384 experiments are all biased too low over both the Pacific and Atlantic. Generally, the correlations are much better below ~6 km for these two cases with the C96 experiment being better.

### 5.3 Height-latitude profile comparisons over the Atlantic during ATom-1

The ATom-1 flight profiles allow a more detailed comparison of aerosol spatial patterns from different aerosol sources with the model. High values of OC and BC from fires were observed on 8/15/16 and 8/17/16 over the Atlantic, as were high values of dust and sea-salt. The flight track of height-latitude profiles of OC, BC and sulfate for these combined days are shown in Figure 20 for the ATom-1 measurements and the 2 model experiments. The model shows very good performance in reproducing the profiles of OC in the two model experiments, especially the biomass burning plumes near the tropics, though the model results show slightly low biases. But they also show some bias for OC at levels above 4~5 km over the north Atlantic, where model results show high biases. Overall, predicted BC (middle column of Figure 20) is able to capture variations of





latitude-height profiles, however they are underpredicted in the biomass burning plumes near the tropics from the surface to 5 km height in both model experiments. Similar to the OC profiles, the model results overpredict above 4-5 km height levels. It appears the model does not reproduce the injection height very well for the biomass burning emissions over this area, possibly due to relative weak convection or a low modeled injection height. This suggests that the injection height of the fire emissions along with the wet removal at higher levels still need to be improved to have better 3-D distributions for biomass burning emissions. For sulfate (right column), the model experiments show high concentrations at low altitude, similar to the observations, though there are still some differences for the plume location at 2-4 km height that shift the plume from near the equator to near 20°N in the model experiments. Over the equatorial areas at about 2-4 km height, the observed sulfate concentration is underestimated by about 30% by modeled results, which may also relate to the injection height of biomass burning that results in much lower $SO_2$ at this altitude since $SO_2$ is one of the important precursors for sulfate production. Meanwhile, the sulfate concentration above 6 km is overestimated over the tropics while underestimated near the surface. The C96 experiment shows higher values near 20°N than that of the C384 experiment, which are also overestimated.

Figure 21 shows the comparisons of the naturally occurring dust and sea salt aerosols for the same time period. In the left column of dust, we also include the NGACv2 results. For more consistent comparisons, here the modeled dust results are summed up by the first two bins to match the observation particle size range (less than 3 $\mu$m). The GEFS-Aerosol predictions show agreement in the dust height-latitude profiles with the observations and the C96 and C384 cases exhibit similar patterns. The observed dust plumes are reproduced by the model over 15-35°N, but the model appears to underestimate wet removal in the upper levels that results in the overestimation of dust above 7-8 km height in northern Atlantic and above 5 km height in the tropical southern Atlantic. On the other hand, the NGACv2 prediction shows a very large bias over broad areas of the north Atlantic and the tropical southern Atlantic. A high dust plume near 35°S has not been captured well by the model from the surface to the upper levels, which may be caused by missing dust events over South America. For sea salt, the two experiments of both C96 and C384 are able to predict patterns consistent with the observation, especially from the surface to about 4-6 km height. The C96 results are much higher than that of the C384 above 4-5 km height, probably due to less wet removal in upper levels in the C96 case.

## 6. Summary and future plan

Since the dynamical core of FV3 developed by GFDL has been selected by NOAA to be the dynamical core for the Next Generation Global Prediction System (NGGPS), development of a coupled weather and atmospheric chemical composition model for chemical weather and air quality forecasting based on the FV3 framework has begun couple of years ago. The development as a single ensemble member of the Global Ensemble Forecast System (GEFS) has been completed. This new model, referred to as GEFS-Aerosols, was implemented as one member of the GEFS into operations as part of NOAA's first coupled UFS model in September 2020 and replaced previous operational global aerosol prediction system NGACv2 at NCEP.





The chemical component of atmospheric composition in GEFS-Aerosols is based on WRF-Chem, which is a community
modeling system used by thousands of users worldwide. The aerosol modules are based on modules from the GOCART model.
Features of the new model include: 1) the biomass burning plume rise module added from WRF-Chem; 2) the FENGSHA
dust scheme implemented and developed by NOAA Air Resources Laboratory (ARL); 3) all sub-grid scale tracer transport
and deposition is handled inside the physics routines requiring consistent implementation of positive definite tracer transport
and wet scavenging in the SAS scheme; 4) the updated background fields of OH, $H_2O_2$ and $NO_3$ from GMI model; 5) biomass-
burning emission calculations based on the GBBEPx V3 emission and FRP provided by NESDIS; and 6 ) global
anthropogenic emission inventories derived from CEDS and HTAP. This new model is able to forecast the higher-resolution
distribution of primary air pollutants of aerosols: black carbon, organic carbon, sulfate, and dust and sea salt each with five
size bins. Meanwhile, it is also capable of handling volcanic eruptions, which can inject vast quantities of particulates into the
atmosphere.
Several sensitivity experiments using different emissions inputs indicate that the model shows the best performance matching
the observations when configured to use the CEDS anthropogenic emission and GBBEPx V3 fire emissions with plume-rise
module. For more extensive evaluation, we performed 9 months of Day-1 real-time forecast of GEFS-Aerosols starting in July
2019 and the predicted AOD was used to compare with the satellite observations from MODIS and VIIRS, reanalysis data of
ICAP-MME and MERRA-2, AERONET observations, and the model predictions from GEOS-5 and NGACv2. Overall,
GEFS-Aerosols is a substantial improvement for both composition and variability of aerosol distributions over those from the
currently operational global aerosol prediction system of NGACv2. Globally, the GEFS-Aerosols predicted biases with respect
to GEOS-5 forecast for dust, OC and sulfate AOD were improved compared to those from NGACv2. Substantial improvements
were seen for the total AOD prediction when compared with MERRA-2 reanalysis during the period of July to November
2019. Though there are still some high biases over southern African fire region and eastern Asia and low biases over south
America and dust source regions, GEFS-Aerosols reproduces the prominent temporal and geographical features of AOD as
represented by satellite observations and reanalysis data, like dust plumes over North Africa and the Arabian Peninsula,
biomass burning plumes in Southern Hemisphere, South America, Northwestern America and Eastern Europe, polluted air
over Eastern and Southern Asia, and high-altitude sea-salt bands. We also sampled the forecast total AOD of GEFS-Aerosols
and NGACv2 in the same location as 60 AERONET sites, which are spread globally and represent different aerosol regimes,
and compared their variations for the 7/5/19-11/30/19. Much higher correlation coefficients against AERONET data are
indicated for the GEFS-Aerosols than those for NGACv2 globally, and are quite comparable to those of the ICAP-MME.
During the biomass burning events, GEFS-Aerosols captured major fires over southern Africa, Siberia, Central Amazon and
Central South America much better than NGACv2. Part of the improvement may be due to the vertical transport by the plume-
rise module. Generally, the total AOD time series of GEFS-Aerosols predictions matches closely to those of ICAP and
AERONET during most of the time from July to November 2019 at the AERONET sites over South America, except there
are some minor under predictions of several highest AOD episodes. In contrast, NGACv2 substantially underpredicted almost





throughout the whole period and almost entirely missed many high AOD events. For the southern African event, the GEFS-Aerosols predictions are able to capture the daily total AOD variations seen in the AERONET observations, even better than that of the ICAP total AOD at the sites near the fire source regions, though there are overpredictions at the sites in downwind areas, which may be due to the lack of removal process or uncertainties of fire emission in central and southern Africa. In contrast, the NGACv2 results show under prediction in total AOD forecast at most of the AERONET sites in this region.

Overall, the model predicts total AOD variation by GEFS-Aerosols indicates much better performance than that of NGACv2 over western North Africa. Although GEFS-Aerosols shows reductions in dust emissions over the Saharan dust source, the correlations with observations from downwind AERONET sites in western Africa are improved over those for NGACv2. The largest biases and discrepancies of GEFS-Aerosols and NGACv2 are both indicated in the sites in Tajikistan, which may be associated with a missing dust source near this site for both models.

We also evaluated predicted aerosols concentrations with different fires emissions and resolution against the ATom-1 aircraft measurements from July to August 2016. Overall, predicted aerosol concentrations are quite comparable to the ATom-1 measurements along the flight tracks globally with no substantial differences for equivalent C384 and C96 experiments. Wet removal appears stronger in C384 than that in C96, which results in less $SO_4$, OC, BC and sea-salt for C384 relative to C96 over some areas. The model shows good performance in reproducing vertical profiles of OC, BC and sulfate, and the location of fire plumes was captured well overall. Sulfate over the Pacific, southern and tropical Atlantic is significantly underpredicted, suggesting an underestimation in the oceanic sulfur sources, such as DMS. A clear trend in increased overprediction with altitude for BC suggests that further refinements in characterizing precipitation scavenging of aerosol in GEFS-Aerosols is needed, since this is the only loss process for BC other than surface deposition.

This paper overviews advances and challenges in model development for operational atmospheric aerosol predictions at NOAA. This implementation advanced the global aerosol forecast capability for NOAA and made a step forward toward developing a global aerosol data assimilation system. Currently, the assimilation of AOD based on satellite observations is under development to constrain aerosol distributions in the GEFS-Aerosols model. Initial testing shows promise for improvement of predictions as well as limitations indicating the need for refinements in quality control, data assimilation impacts on aerosol composition and vertical distribution, as well as a need for bias correction of satellite observations. With bias and other errors substantially reduced in the GEFS-Aerosols, especially when it is equipped with an aerosol data assimilation system, the model provides a good starting point from which to investigate at the impact on weather predictions out to sub-seasonal and seasonal scales when including the aerosol feedbacks on atmospheric system.

**Code and data availability**

The GEFS-Aerosols v1 code and model configuration for aerosol forecast here are available at https://doi.org/10.5281/zenodo.5655290. ATom-1 data is publicly available at the Oak Ridge National Laboratory Distributed Active Archive Center: https://daac.ornl.gov/ATOM/guides/ATom_merge.html (Wofsy et al., 2018).



# 1 Author contribution

Li Zhang and Raffaele Montuoro were the major developers of the GEFS-Aerosols model, including implementing and coupling the aerosol components to the FV3GFSv15 meteorological model. Stuart McKeen helped to process the anthropogenic emission and background input data, provided suggestions during the development of GEFS-Aerosols, and evaluated the model performance with ATom observations. Barry Baker developed and implemented the FENGSHA dust scheme into GEFS-Aerosols. Partha Bhattacharjee helped to evaluate the GEFS-Aerosols real-time and operational predictions. Georg Grell provided oversight of the model development. Li Zhang and Judy Henderson developed the workflow for GEFS-Aerosols prediction and worked with Li Pan to perform and manage the real-time and retrospective forecasts. R. Ahmadov provided guidance on the implementation of the fire plume rise scheme. Shobha Kondragunta, Xiaoyang Zhang, and Fangjun Li provided the GBBEPx V3 data. The other co-authors provided help, suggestions and project management throughout the GEFS-Aerosols modeling system development.

# 12 Competing interests

The authors declare that they have no conflict of interest.

# 14 Acknowledgements

Li Zhang, Raffaele Montuoro, Haiqin Li and Ravan Ahmadov were supported by funding from NOAA GSL Award Number NA17OAR4320101. This work was also supported by the UFS Research to Operations Medium Range Weather/Seasonal to Subseasonal Atmospheric Composition sub-project.

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

**Table.**
**Table 1.** AERONET site information.

| Stations Numbers | Station Names | (Latitude, Longitude) |
|---|---|---|
| 1 | Dakar | (14.39N, 16.95W) |
| 2 | Cape Verde | (16.73N, 22.93W) |
| 3 | Banizoumbu | (13.54N, 2.66E) |
| 4 | Tamanrassett | (22.79N, 5.53E) |
| 5 | Tenerife | (28.47N, 16.24W) |
| 6 | Saada | (31.62N, 8.15W) |
| 7 | Ben Salem | (35.55N, 9.91E) |
| 8 | Sede Boker | (30.85N, 34.78E) |
| 9 | Dewa | (24.76N, 55.36E) |
| 10 | Granada | (37.16N, 3.60W) |
| 11 | Cape San Juan | (18.38N, 65.62W) |
| 12 | Dushanbe | (38.55N, 68.85E) |
| 13 | Dalanzadgaad | (43.57N,104.41E) |
| 14 | Bejing | (39.97N,116.38E) |
| 15 | Kanpur | (26.51N, 80.23E) |
| 16 | Kyiv | (50.36N, 30.49E) |
| 17 | Barcelona | (41.38N, 2.11E) |
| 18 | Leipzig | (51.35N, 12.43E) |
| 19 | Sochengcho | (37.42N,124.73E) |



| 20 | Singapore | (1.29N,103.78E) |
|---|---|---|
| 21 | Reunion St Denis | (20.90S, 55.48E) |
| 22 | Lumbini | (27.50N, 83.28E) |
| 23 | Cape Fuguei | (25.29N,121.53E) |
| 24 | Lake Argyle | (16.10S,128.74E) |
| 25 | Chilbolton | (51.14N, 1.43W) |
| 26 | Opal | (79.99N, 85.93W) |
| 27 | Resolute Bay | (74.70N, 94.96W) |
| 28 | Thule | (76.51N, 68.76W) |
| 29 | Kangerlussuaq | (66.99N, 50.62W) |
| 30 | Tomsk | ( 56.45N, 85.04E) |
| 31 | Hornsund | (77.0N, 15.54E) |
| 32 | Alta Floresta | (9.87S, 56.10W) |
| 33 | Santa Cruz Utepsa | (17.76S, 63.20W) |
| 34 | Itajuba | (22.41S, 45.45W) |
| 35 | La Paz | (16.53S, 68.06W) |
| 36 | SEGS Lope Gabon | (0.202S, 11.60E) |
| 37 | Ascension Island | (7.97S, 14.41W) |
| 38 | Bamenda | (5.94N, 10.15E) |
| 39 | Mongu Inn | (15.26S, 23.13E) |
| 40 | Misamfu | (10.17S, 31.22E) |
| 41 | Maun Tower | (19.9S, 23.55E) |
| 42 | Windpoort | (19.36S, 15.48E) |
| 43 | Lubango | (14.95S, 13.44E) |
| 44 | Bonanza Creek | (64.74N,148.31W) |
| 45 | Fort McMurray | (56.75N,111.47W) |
| 46 | Chapais | (49.82N, 74.97W) |
| 47 | Saturn Island | (48.77N,123.12W) |
| 48 | Missoula | (46.91N,114.08W) |
| 49 | Camaguey | (21.42N, 77.85W) |
| 50 | Neon Wood | (47.12N, 99.24W) |
| 51 | GSFC | (38.99N, 76.84W) |
| 52 | Monterey | (36.59N,121.85W) |
| 53 | Toronto | (43.79N, 79.47W) |
| 54 | Red Mountain Pass | (37.90N,107.71W) |
| 55 | Tucson | (32.23N,110.95W) |
| 56 | Appalachian State | (36.21N, 81.69W) |
| 57 | Cartel | (45.38N, 71.93W) |
| 58 | Mauna Loa | (19.53N,155.57W) |
| 59 | ARM SGP | (36.6N, 97.48W) |
| 60 | Univ. of Wisconsin | (43.07N, 89.41W) |






**Table 2.** ATom-1 column sum statistics of mean bias and correlation for sulfate, OC, EC and dust.

| Species | N | Obs. median (mg/m²) | FV3-C96 MMO | FV3-C384 MMO | NGAC-v2 MMO | FV3-C96 r-coeff. | FV3-C384 r-coeff. | NGAC-v2 r-coeff. |
|---|---|---|---|---|---|---|---|---|
| Sulfate | 153 | 0.58 | 0.82 | 0.72 | - | 0.70 | 0.63 | - |
| Organic Carbon | 146 | 0.55 | 1.38 | 1.03 | - | 0.85 | 0.80 | - |
| Elemental Carbon | 152 | 0.011 | 4.54 | 3.35 | - | 0.80 | 0.78 | - |
| Dust (< 3. µm diam) | 130 | 0.038 | 0.47 | 0.54 | 46.37 | 0.39 | 0.39 | 0.39 |

N is the sample number, MMO stands for Median Model to Observed Ratio, and r-coeff. is the Pearson correlation r-
coefficient.
**Figure Captions.**
**Figure 1.** Diagram showing the GSDCHEM component within the NEMS infrastructure.
**Figure 2.** (a) Diagram of FV3GFS-GSDCHEM coupled structure; (b) Flowchart of GEFS-Aerosols forecast workflow.
**Figure 3.** Anthropogenic emissions between CEDS and HTAP for $SO_2$ (mole/km²/hr), BC and OC (ng/m²/s) on summer of
July.
**Figure 4.** OC fire emissions of MODIS and GBBEPx V3.
**Figure 5.** Total AOD from a GEFS-Aerosols 18-hour forecast using three different fire emission schemes and their
comparisons with 18-hr forecasts by ICAP and NGAC on June 30, 2019.
**Figure 6.** Total AOD forecast on 25th August verified against the NGAC model, MERRA2 reanalysis data and satellite
observations of VIIRS and MODIS. The 18z forecasts from both models for that day and daily satellite data are used in the
figure. Satellite data gaps are in white.
**Figure 7.** Day 1 AOD prediction of GEFS-Aerosols and NGAC compared with MERRA-2 reanalysis and MODIS averaged
during 7/5/19-11/30/19.
**Figure 8.** Day 1 AOD forecast biases of GEFS-Aerosols and NGAC compared with GEOS-5 averaged during 7/5/19-11/30/19
for dust, OC and sulfate.
**Figure 9.** Differences of GEFS-Aerosols and NGAC Day 1 predictions of total AOD compared with MERRA-2 reanalysis
averaged during 7/5/19-11/30/19.
**Figure 10.** Correlation coefficients between AERONET daily total AOD observations and model predictions by GEFS-
Aerosols, ICAP, and NGAC for the period 7/5/19-11/30/19. Correlation coefficients are at the 95% confidence interval.
**Figure 11.** Day 1 AOD forecasts of GEFS-Aerosols, ICAP, and NGAC verified against AERONET sites in South America
during 7/5/19-11/30/19.





**Figure 12.** Day 1 AOD forecasts of GEFS-Aerosols, ICAP, and NGAC verified against AERONET sites in Africa during
2  7/5/19-11/30/19.

**Figure 13.** Day 1 AOD forecasts of GEFS-Aerosols, ICAP, and NGAC verified against AERONET sites in dust source regions
and surrounding downwind areas during 7/5/19-11/30/19.
**Figure 14.** Daily AERONET total AOD versus modeled total AOD from GEFS-Aerosols (blue) and NGAC (orange) at the
AERONET sites of (a) Tamanrassett, (b) Cape Verde, and (c) Tenerife with linear regression fits.
**Figure 15.** GEFS-Aerosols and NGAC day 1 total AOD forecast time series against MERRA-2 reanalysis data averaged over
major global regions of North Africa (0°-35°N, 18°W-30°E), North Atlantic Ocean, (0°-40°N, 10°-80°W), Southern Africa
(0°-35°S, 8°-35°E), South Atlantic (0°-35°S, 40°W-20°E), South America (0°-35°S, 35°W-80°W), Europe (35°-65°N, 10°W-
50°E), East Asia (20°-48°N, 100°-140°E), Eastern USA (25°-48°N, 68°-95°W), and Western USA (25°- 48°N, 95°-125°W).
**Figure 16.** Tropospheric column sums of OC for (a) NASA DC-8 observations; (b) GEFS-Aerosols at C384 resolution using
GBBEPx v3 fire emissions, and (c) GEFS-Aerosols at C96 resolution using GBBEPx v3 fire emissions.
**Figure 17.** Tropospheric column sums of dust for (a) the NASA DC-8 observations; (b) GEFS-Aerosols at C96 resolution; (c)
GEFS-Aerosols at C384 resolution; and (d) NGAC.
**Figure 18.** Vertically resolved statistical comparisons of OC, BC and sulfate for the DC-8 flight tracks over the Pacific (July
29-August 8) and Atlantic (August 15-23).
**Figure 19.** Vertically resolved statistical comparisons of dust and sea salt for the DC-8 flight tracks over the Pacific (July 29-
August 8) and Atlantic (August 15-23).
**Figure 20.** Height-latitude profiles of OC, EC and sulfate over Atlantic on August 15 and 17th, 2016 for (a) the ATom-1 DC-
8 observations; (b) GEFS-Aerosols at C384; and (c) GEFS-Aerosols at C96.
**Figure 21.** Height-latitude profiles of dust and sea salt over Atlantic on August 15 and 17th, 2016 for (a) the ATom-1 DC-8
observations; (b) GEFS-Aerosols at C384; (c) GEFS-Aerosols at C96; and (d) NGAC.







Figure 1. Diagram showing the GSDCHEM component within the NEMS infrastructure.



Figure 2. (a) Diagram of FV3GFS-GSDCHEM coupled structure; (b) Flowchart of GEFS-Aerosols forecast workflow.







Figure 3. Anthropogenic emissions between CEDS and HTAP for SO2 (mole/km2/hr), BC and OC (ng/m2/s) on summer of July.





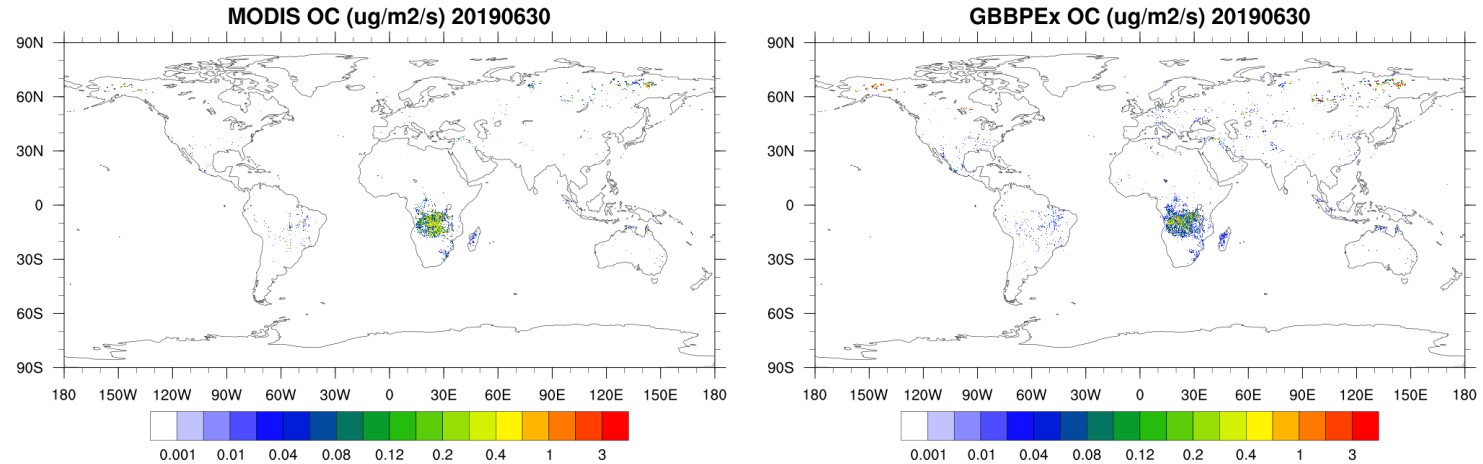

Figure 4. OC fire emissions of MODIS and GBBEPx V3.





Figure 5. Total AOD from a GEFS-Aerosols 18-hour forecast using three different fire emission schemes and their comparisons with 18-hr forecasts by ICAP and NGAC on June 30, 2019.







Figure 6. Total AOD forecast on 25th August verified against the NGAC model, MERRA2 reanalysis data and satellite observations of VIIRS and MODIS. The 18z forecasts from both models for that day and daily satellite data are used in the figure. Satellite data gaps are in white.





Figure 7. Day 1 AOD prediction of GEFS-Aerosols and NGAC compared with MERRA-2 reanalysis and MODIS averaged during 7/5/19-11/30/19.



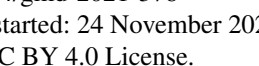

Figure 8. Day 1 AOD forecast biases of GEFS-Aerosols and NGAC compared with GEOS-5 averaged during 7/5/19-11/30/19 for dust, OC and sulfate.



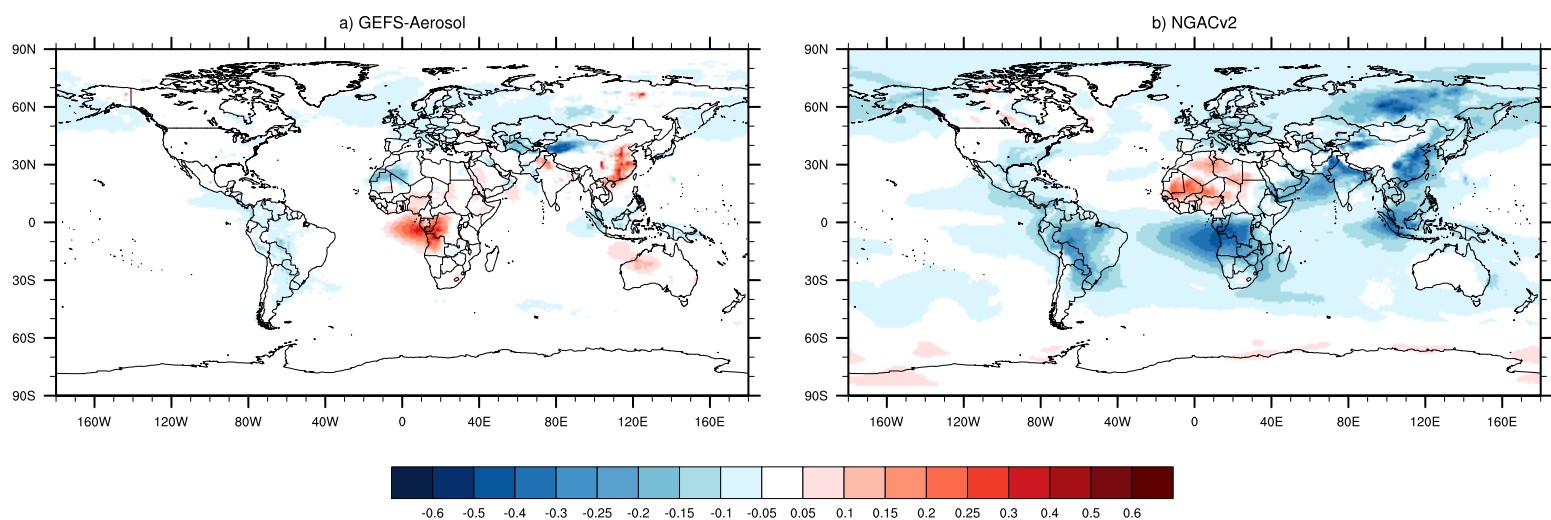

Figure 9. Differences of GEFS-Aerosols and NGAC Day 1 predictions of total AOD compared with MERRA-2 reanalysis averaged during 7/5/19-11/30/19.



Figure 10. Correlation coefficients between AERONET daily total AOD observations and model predictions by GEFS-Aerosols, ICAP, and NGAC for the period 7/5/19-11/30/19. Correlation coefficients are at the 95% confidence interval.





Figure 11. Day 1 AOD forecasts of GEFS-Aerosols, ICAP, and NGAC verified against AERONET sites in South America during 7/5/19-11/30/19.







Figure 12. Day 1 AOD forecasts of GEFS-Aerosols, ICAP, and NGAC verified against AERONET sites in Africa during 7/5/19-11/30/19.



Figure 13. Day 1 AOD forecasts of GEFS-Aerosols, ICAP, and NGAC verified against AERONET sites in dust source regions and surrounding downwind areas during 7/5/19-11/30/19.





Figure 14. Daily AERONET total AOD versus modeled total AOD from GEFS-Aerosols (blue) and NGACv2 (orange) at the AERONET sites of (a) Tamanrassett, (b) Cape Verde, and (c) Tenerife with l inear regression fits.





Figure 15. GEFS-Aerosols and NGACv2 day 1 total AOD forecast time series against MERRA-2 reanalysis data averaged over major global regions of North Africa (0°-35°N, 18°W-30°E), North Atlantic Ocean, (0°-40°N, 10°-80°W), Southern Africa (0°-35°S, 8°-35°E), South Atlantic (0°-35°S, 40°W-20°E), South America (0°-35°S, 35°W-80°W), Europe (35°-65°N, 10°W-50°E), East Asia (20°-48°N, 100°-140°E), Eastern USA (25°-48°N, 68°-95°W), and Western USA (25°- 48°N, 95°-125°W).



Figure 16. Tropospheric column sums of OC for (a) NASA DC-8 observations; (b) GEFS-Aerosols at C384 resolution using GBBEPx v3 fire emissions, and (c) GEFS-Aerosols at C96 resolution using GBBEPx v3 fire emissions.



Figure 17. Tropospheric column sums of dust for (a) the NASA DC-8 observations; (b) GEFS-Aerosols at C96 resolution; (c) GEFS-Aerosols at C384 resolution; and (d) NGAC.



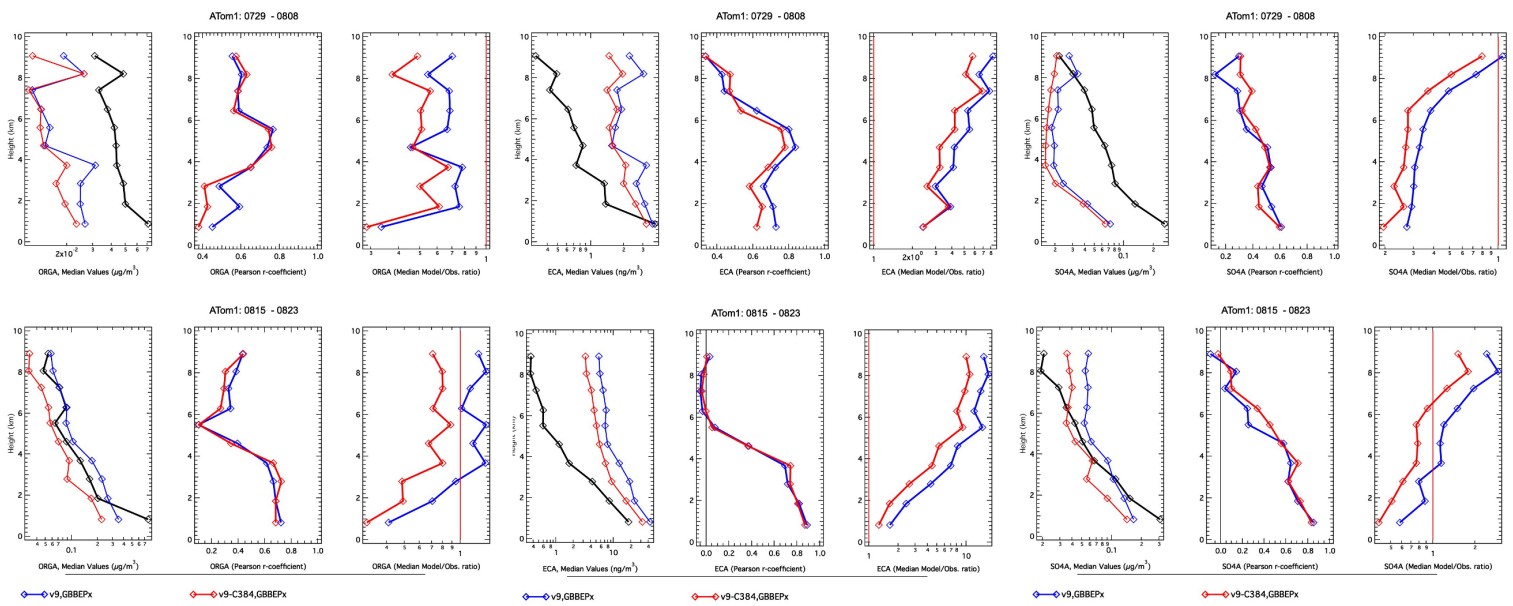

Figure 18. Vertically resolved statistical comparisons of OC, BC and sulfate for the DC-8 flight tracks over the Pacific (July 29-August 8) and Atlantic (August 15-23).





Figure 19. Vertically resolved statistical comparisons of dust and sea salt for the DC-8 flight tracks over the Pacific (July 29-August 8) and Atlantic (August 15-23).







Figure 20. Height-latitude profiles of OC, EC and sulfate over Atlantic on August 15 and 17th, 2016 for (a) the ATom-1 DC-8 observations; (b) GEFS-Aerosols at C384; and (c) GEFS-Aerosol at C96.





Figure 21. Height-latitude profiles of dust and sea salt over Atlantic on August 15 and 17th, 2016 for
(a) the ATom-1 DC-8 observations; (b) GEFS-Aerosols at C384; (c) GEFS-Aerosol at C96; and (d) NGACv2.