# Peer review of "Development and Evaluation of the Aerosol Forecast Member in"

_Geoscientific Model Development, 2021_

## Author Comment (AC1)

*Reply to anonymous referee #1*

**Overview**

The paper gives a model description and presents the evaluation results of the aerosol forecast of the GEFS-Aerosols v1 system. This system is a newly developed aerosol module coupled on-line to NOAA's FV3 Global Forecast System (FV3GFS) by means of the National Unified Operational Prediction Capability (NUOPC). The evaluation results are compared against the performance of the previous NGAC v2 aerosol forecast system showing a clear improvement in many aspects of the aerosol forecast.

Reply: We really appreciate the reviewer's very helpful comments and suggestions. The paper has been revised throughout based on all the general and specific comments listed by the reviewers, including the text, figures, references, etc.

**General remarks**

The paper inter-compares several aerosol model/analysis products (ICAP, GEOS5, MERRA, NGAC) with the GFES Aerosol forecast results. However, there is no stringent approach to the choice of these data sets for the different aspects. This makes the paper appear somewhat convoluted and too long. I recommend focusing on the forecast by GEFS-Aerosols v1 and its predecessor NGAC v2 only throughout the paper. These two data sets should be evaluated against observations and observation-based re-analysis data sets such as MERRA. The evaluation results of the two systems should be intercompared for all the discussed topics. If the authors still which to include other forecast or model data sets (ICAP, GEOS5) they need to describe these modelling systems in such a way that the identified differences in the evaluation against observations and observation-based reanalyses can be explained. There is little value in pointing out that GFES Aerosol is higher or lower than ICAP or GEOS5 without saying which one is better, i.e. closer to the observations.

Reply: We thank the reviewer's very good suggestion. We have removed the evaluation using GEOS-5. We keep the ICAP analysis data for evaluation because the MARRA-2 reanalysis data is not real-time, which has ~1-2 month time lag with respect to real-time. While in our real-time or operational forecast, the ICAP analysis is able to provide synchronous comparison. Before choosing the ICAP analysis data, we have done some evaluation with other observation (satellite and ANERONET observation), including the correlation and RMSE between ICAP and AERONET (see Table 1 some descriptions in Section 4), it obviously that the ICAP analysis has the highest correlation and smallest RMSE with respect to AERONET. It suggests that the ICAP analysis is quite close to the observation and is good to use it as the global evaluation data when MERRA-2 is not available in the real-time or operational forecast. We have emphasized it in the Section 2.2.

The paper remains too in explaining the reasons for the difference in the evaluation results between GEFS-Aerosols v1 and NGAC v2. It should be stated more clearly what aspects (emissions, removal processes, aerosol conversion, resolution, transport etc.) is assumed to be the reason for the mainly improved performance of the newer system. Further, I strongly recommend adding a table that summaries the communalities and differences between GEFS-Aerosols v1 and NGAC v2 as the reader is not made familiar with the configuration of NGAC v2.

Reply: We thank the reviewer's comments. As we introduced in Section 1 and Section 2.2.3, the NGACv2 is the previous global aerosol forecast system in NCEP. The major differences between

GEFS-Aerosols model and NGACv2 is not only the chemical model part, the atmospheric/weather model is completely different. NGACv2 is implemented into the NOAA Environmental Modeling System (NEMS) global spectral model (GSM), which has been developed at NCEP to implement the standalone global forecast system (GFS) in the NEMS framework in 2006 (Black et al., 2007, 2009). While GEFS-Aerosols is implemented into the global Finite-Volume cubed-sphere dynamical core (FV3) developed by GFDL within the physical scheme of GFSv15. There are a lot of differences in these two atmospheric models, including dynamical core, resolution, physics and microphysics, land surface model, etc. It is hard to exactly quantify the where the improvement may come from. In addition, the evaluation of the NGAC v2 model has been published in (Wang et al., 2018 and Bhattacharjee et al., 2018), which is not the scope of current paper. The comparisons between GEFS-Aerosols and NGACv2 in this study are only the evidence to show the improvement for replacement.

Taking the very good suggestions from the reviewer, we have added a table to summarize the comparison of model configurations between GEFS-Aerosols v1 and NGACv2 in Table 2, the NGACv2 information is from Wang et al., 2018.

The evaluation of the forecast consists mainly of the comparisons with respect to observations and analyses of total or speciated AOD. It is an omission of the paper that routine surface PM observations are not included in the evaluation. PM 2.5 observation data sets are widely available, and the forecast of surface PM should be a main objective of any state-of-the-art aerosol forecasting system.

Reply: We thank the reviewer's suggestion. As the first paper to introduce the operational GEFS-Aerosols, it can not include all the evaluations, especially the $PM_{2.5}$, which need more detailed analysis related to local availability of observation data, emission, topography, and weather. We have the other groups working on evaluating different aspect of the model performance, including the surface $PM_{2.5}$ for GEFS-Aerosols prediction with more detailed validation against the Open-AQ $PM_{2.5}$. They are preparing the drafts and will submit them as separate papers soon.

Bhattacharjee, P. S. ; Zhang, L.; Baker, B.; Pan, L., Mountuoro,R., Grell, G. and McQueen, J : Evaluation of Aerosol optical depth forecast and surface PM2.5 from NOAA's Global Aerosol Forecast Model (GEFS-Aerosols), 2022, to be submitted.

Jeong, G.-R., B. Baker, P. C. Campbell, R. Saylor, and Partha Bhattacharjee, Daniel Tong, and Youhua Tang: Updating Anthropogenic Emissions in NOAA's Global Ensemble Forecast System with Aerosols (GEFS-Aerosols): Application of a Bias-Scaling Method. 2022, to be submitted.

The paper shows detailed comparisons against speciated AOD (BC, OC, SO4/SO2). However, the speciated AOD are model results, i.e. not provided by observation instruments, which mainly observe/retrieve total AOD. Even data assimilation of these observations for the re-analysis (MERRA) is no guarantee that the speciation of the reanalysis is better than the modelled speciation. Therefore, the evaluation with total AOD observations (AERONET) should be given a much larger emphasis in the paper. It is urgently recommended to also include the biases or RMSE (and not only the correlation) against AERONET observations in the paper. At the same time, the applied optimisations of the AOD calculation to account for aerosol species (Nitrates, SOA) not modelled by GEFS-Aerosols needs to be better explained.

Reply: We completely agree with reviewer's comment that the speciated AOD from either MERRA-2 or GEOS-5 has model dependency, and it may not be accurate enough, especially both MERRA-2 and GEOS-5 include data assimilation. As the reviewer's suggestions, we have added the RMSE for AERONET in Table 1, and also calculate the RMSE for Figure 5 in the revised manuscript, and corresponding descriptions in Section 4.2, and 4.3. We have also explained the AOD calculation to account for the absence other aerosols in GOCART aerosol scheme in Section 2.1.3.

The paper also includes and evaluation with flight campaign data (ATOM-1). While this is an interesting aspect of the scientific verification, it seems inconsistent that this section includes a discussion of the impact of spatial resolution which is not discussed before and which is not very large. In the interest of keeping the paper short, I would omit the resolution discussions.
Reply: We thank the reviewer's suggestions. We have removed the discussion of C96 resolution in Section 5.

Finally, the paper requires more clarification of the implied benefits of aerosol – weather feedbacks and the relation of this aerosol-aware forecast as part of the NWP ensemble of the NOAA Environmental Modeling System. It remains unclear what benefits were achieved by including the aerosol ensemble member. If no results can be presented as part of the paper, this should be stated more clearly (also in the title) and less emphasis should be given on weather-composition feedbacks as part of the introduction.
Reply: We thank the reviewer's comments. Currently, the aerosol (from the chemical model) feedback on the atmospheric model has not yet been included, which is next step in our plan. We have clarified that in Section 1:" There is not aerosol feedback on the atmospheric model of GEFS, and the aerosols are not in any way interactive with the radiation and clouds". Also, in the Global Ensemble Forecast System, there is no aerosol ensemble member, the ensemble members are with respective to the weather model for separate forecasts. The GEFS-Aerosols is only using one of the same weather model as other GEFS members but includes the prognostic aerosols. We have described and clarified it in Section 2.1.1 as: "The GEFS is a weather forecast modeling system made up of 31 separate forecasts, or ensemble members, which have the same horizontal (~25 km) and vertical resolution (64 layers from the surface to 0.2 hPa). The GEFS-Aerosols is only using one of the same weather model as other GEFS members but includes the prognostic aerosols. It is about 2.5 times computational cost when include the aerosol component in the forecast. In the operation, there is no execution time since the it only performs 120 hours forecast with aerosol component, while other members without aerosol component would perform 384 hours forecast. The NCEP started the GEFS to address the nature of uncertainty in weather observations that are used to initialize weather forecast models and uncertainties in model representation of atmospheric dynamics and physics. The aerosol component coupled with FV3GFS v15 has been merged into the GEFS, as a single ensemble member named GEFS-Aerosols, for real-time and retrospective forecast that preceded operational implementation, which occurred in September 2020."

**Specific comments:**
Abstract:
L 11: no need to include references and mentioning of FIM-Chem in the abstract.
Reply: Revised.

L 22: Please mention the main reasons for the improvements in the abstract
Reply: Revised.

P 3 L 10 -22 : The discussion of the various feedbacks would only be justified if the paper reports about modified NWP results because of considering aerosol-weather feedback. This seems not the case and the text should be shortened substantially.
Reply: The aerosol feedback is an important part in NWP and it is planning in next step implementation, which need to be based on current model development and performance in AOD forecast. We have shortened and revised this part to better describe this motivation.

P 4 L 22: Here or elsewhere add the spatial resolution of the NRT GEFS-Aerosols v1 forecasts
Reply: Added.

P 5 L15: Please clarify how the emissions are added and how this is linked to the diffusion and convection tracer transport parameterizations.
Reply: Revised. The emissions are no added into the FV3 atmospheric model of the physical part, however, the emissions are added into the aerosol concentration in the chemical model. Then the chemical tracers are passed back to atmospheric model for transport and advection after the chemical related processes. In the physical part (GFS scheme) of the FV3 atmospheric model, all sub-grid scale transport and convective deposition is handled inside the atmospheric physics routines of simplified Arakawa–Schubert (SAS) scheme. All the chemical tracer related processes, such as emission, dry deposition, settling, large-scale wet deposition, chemical reactions are handled by the chemical model. We have clarified it Section 2.1.1 and Section 2.1.2.

P 5 L 17: Please expand on why wet deposition by large scale and convective precipitation is dealt with in different components.
Reply: We thank the reviewer brought up this important point. The most consistent way would be to have all wet deposition done inside the physics. This was easily possible for the convective parameterization. However, moving the large-scale wet deposition into the microphysics routine was at this point not an option, but it will be done at a later stage of our plan for future development. It is a non-trivial task to include wet scavenging and possibly aqueous phase chemistry in the explicit microphysics scheme. An additional complication may be that the microphysics parameterization might be switched with the next implementation.

P 5 L 17: Please comment in this section about the consistency of land use and other climatological surface fields (z0, vegetation type etc.) between the dynamical core and the aerosol model.
Reply: Revised in Section 2.1.2. All the Metrological fields (such as land use and other climatological surface fields (z0, vegetation type etc.) are imported from the FV3 atmospheric model to the chemical model to drive the aerosols components. So there is no inconsistency.

P 5 L 21: What is the motivation to include FIM-chem here?
Reply: Though the current atmospheric composition option in the GEFS-Aerosols model is based on the simple bulk aerosol modules from WRF-Chem, however WRF-Chem is a reginal model. While the first time to include the same aerosol component into global model is in FIM-Chem

model, which showed good performance. That is one the reasons we chose it for the global aerosol forecast of GEFS-Aerosols.

P 5 L 24: Please provide more details on the oxidant fields. Are these statistic climatologies or do they change in space and time because of advection. Is SO2 a tracer?
Reply: Provided in Section 2.1.2. The GOCART model background fields of prescribed OH, $H_2O_2$, and $NO_3$ have been replaced by the newer version of 2015 from the NASA GEOS Global Modeling Initiative (GMI) Chemical transport model (https://acd-ext.gsfc.nasa.gov/Projects/GEOSCCM/MERRA2GMI/). These are monthly mean data and these prescribed OH, $H_2O_2$, and $NO_3$ fields would not be transported and changed in space. They are provided to the calculation in chemical reactions for gaseous sulfur oxidations. They will not be passed back to FV3 for transport and advection, no loss and sink. Yes, SO2 is a transport chemical tracer in the model.

P 6 L 11: It is not clear from the text what the threshold values are based on … wind tunnel experiments ?
Reply: Revised. The threshold friction velocities are based on wind tunnel measurements done in both the laboratory and field (Gillette et al., 1980).

P 6 L 18: please add (BSM)
Reply: Added.

P 6 L24: What is a 3-year climatology?
Reply: Revised. The 3-year climatology refers to a monthly average over 3 observation years, in this case 2016, 2017 and 2018 as these were the latest full years at the time of model development. For example, January would be the average values of the BSM over January of 2016, 2017, and 2018.

P 6 L 25: This section describes more than the coupler – so consider renaming that section or introduce sub-sections.
Reply: Revised.

P 7: A reference to Fig 2a is missing in that section.
Reply: Added.

P 7 L 11: Please indicate the computational cost of the aerosol module in relation to the cost of the dynamical core.
Reply: Added in Section 2.1.3. In operation, the computational cost with aerosol component would take 129 mins for 120 hours forecast using 330 tasks. The atmospheric model without aerosol component run would take 168 mins for 384 hours forecast using 320 tasks. Therefore, the efficiency for the former is 120/(330x129)=2.82x10-e3 hour/mins, while the latter is 384/(320x168)=7.14e-3hour/min. It is about 2.53 times computational cost when include the chemical model in the forecast.

P 7 L 12: Please indicate the resolution of the 31 non-aerosol members and the resolution of the aerosol member. Are they the same? How does the potentially increased cost and execution time of the aerosol member impact the execution time of the ensemble as a whole?

Reply: The resolution of the 31 non-aerosols members and the aerosols member are the same. The aerosol member only performs 120 hours forecast, however, the non-aerosol members perform 384 hours forecast, so the no execution time issue because the aerosol member finishes the forecast before other non-aerosol members, so there is no execution time issue.

P 7 L 19: Fig 2b is not clear at all. The names of specific routines such as checkic is not of interest for the reader. Why is re-gridding needed if the aerosol module runs at the same resolution as the core? What are the meaning of the green and yellow boxes. How is Fig 2.b related to Fig 2.a.

Reply: All the red abbreviations (e.g. "checkgdas", "gfsgetic" ect.) are the names of the tasks in the xml file of global workflow (https://github.com/NOAA-EMC/global-workflow) to run the GEFS-Aerosols for operational forecast way, which are a uniform way in the operational system (not only for GEFS-Aerosols model forecast) and named by the global workflow designers. The black statements below the red abbreviations are the explanations.  The regriding in the "regrid" step is not used to regrid the meteorological fields from atmospheric model to chemical model, it is not necessary to do that because they are in the same resolution. The "regrid" step is used to regrid the meteorological fields from GFS/GDAS data assimilation system (normally this data is at very high resolution, ~3km) as initial condition (ICs) input to drive the FV3GFS model. The yellow box includes the necessary steps for atmospheric model, while the green box includes the necessary steps for chemical model. We have modified the Fig. 2b to make it more clearly. Fig. 2a is the model coupled structure of GEFS-Aerosols. Fig. 2b shows the steps about how to run the GEFS-Aerosols in operational forecast system using global workflow, including all the tasks of preprocessing (prepare input data before model forecast) and postprocessing (process output data after model forecast), the whole processes are controlled by the global workflow shown as Fig. 2b. We have clarified it in Section 2.1.3.

P 7 L26: As the AOD evaluation is an important aspect of the paper, more detail (here or elsewhere) needs to be provided to understand the impact of the optimization of the AOD calculation on the evaluation results.

Reply: Revised as "The AOD is calculated in the post-processing part of the workflow, using a look-up table (LUT) of aerosol optical properties from NASA GOCART model (Colarco et al. 2010, Colarco et al. 2014), which was implemented in the Unified Post Processor (UPP, https://dtcenter.org/community-code/unified-post-processor-upp). It should be noted that the LUT reflects the impacts of a larger number of aerosol species in the atmosphere than the simple GOCART module treats. Also, considering the bulk aerosol scheme in GOCART, there is no size distribution for OC, BC and sulfate, the LUT may have uncertainties in the AOD calculation. Based on observational validation, some adjustments with factor of 2 have been applied in into LUT calculation to compensate the contributions for the absence of nitrate, ammonium and secondary organic aerosol (SOA) in GOCART."

P 7 L29: This section should be re-arranged to clarify in a better way what the reference data sets are (observations, re-analysis) and what the evaluated forecasts are (GEFS-Aerosols v1 and NGAC v2 and perhaps GEOS5 and ICAP)

Reply: We have re-arranged the section 2.2.

Reply: Table 1 lists number of stations, their location in terms of latitude and longitude. The stations are selected based on years in service and geographic location near the aerosol source regions. Stations covered major aerosol sources: African Dust, Southern Africa and South America (major forest fire stations), mixed aerosol regimes (urban areas in Europe, Asia and N. America), high latitude stations (capture major transport of forest fires from Siberia and Canada). We have updated in Section 2.2.2.

Reply: Added. We have used Collection 6.1 MODIS AOD at 550nm, which has Expected Errors (EEs) of $[ \pm (0.05 + 15\%AOD)]$ and $[ \pm (0.03 + 5\%AOD)]$ for Dark Target retrievals at a 10-km resolution over land and ocean, respectively. The EEs are approximately $[ \pm (0.03+21\%AOD]$ for 'arid' and $[ \pm (0.03+18\%AOD)]$ for 'vegetated' path Deep Blue retrievals at a 10-km resolution over land (Levy et la., 2013).

Reply: According to the reviewer's comment in the general part, we have removed the evaluation using GEOS-5.
Yes. There is data assimilation applied in GEOS5. GEOS-5 Data Assimilation System (GEOS-5 DAS) integrates the GEOS-5 Atmospheric Global Climate Model (GEOS-5 AGCM) with the Gridpoint Statistical Interpolation (GSI) atmospheric analysis developed jointly with NOAA/NCEP/EMC.

Reply: We have revised the ATOM-1 data descriptions and shorten it.

Reply: We have added the quantified numbers of global total emission in the comparison of CEDS and HTAP 2 emissions in Fig. 3. We also added comments about the data represent different reference years and its impact of using the data for simulations in 2019.

P 11 L 9: The section 3.3 remains a bit anecdotic because only plots for selected days are shown. The paper could work without that section and it would just be enough to mention the selected biomass burning data set and injection option. If this section should remain it will require to quantify the mean aerosol biomass burning emissions for the period and to present an evaluation with independent data for the whole period. The comparison with other model and analysis data sets will require a discussion of the underlying vegetation fire data sets, in particular for NCAG v2, which does not seem to capture the fire events.

Reply: We thank the review's good suggestion. We have removed the fire emission comparison section 3.3 and Figure 4-5 in previous manusrcipt.

P 13 L 14: It is not possible to conclude from a map that the temporal variability was captured.
Reply: Revised.

P 13 L 21: Please provide the reasons for that underestimation by NGAC v2.
Reply: The NGAC v2 model is not the scope of this paper and project. We are not the major developers of the NGACv2 (more than 5 years ago) and hard to conclude the underpredicted reasons of NGACv2. It quires further studies and tons of sensitivity experiments to dig into the NGACv2 performance which is out of the current study of GEFS-Aerosols model development. We just got the NGACv2 history output data from NCEP for comparisons. More details about the NGAC v2 model performance and evaluation can be found in Wang et al., 2018 and Bhattacharjee et al., 2018.

P 13 L 22: Please clarify if the GEOS5 is a forecast or an analysis (data assimilation of AOD).
Reply: According to the reviewer's comment in the general part, we have removed the evaluation using GEOS-5. The reanalysis data of AOD are all use MERRA-2 now.

P 14 L 13: The comparison with AERONET AOD is more important for the reader than the inter-comparison of various modelled and analysis data sets. The section should therefore start best with the AERONET comparison.
Reply: We thank the reviewer's suggestion. We have revised this section and started with the AERONET comparison.

P 14 L 29: Please discuss the biases against AERONET and not only the correlations. Please add a figure for the biases (or RMSE) similar to Fig 10 for the correlation.
Reply: We have added a figure for the biases of RMSE similar to Fig.5 other than the correlation. We also add the RMSE values in Table 1.

P 15 L 11: Please motivate the choice of the selected stations. Why were no North-American or Siberian fire events selected?
Reply: Sites are selected based on the following two factors: 1) observation data availability for the duration of the study; 2) Sites that hold long records based on various previous studies.
In table 1 of the AERONENT sites, the Site 30 Tomsk is the only site located close to Siberian fire and only cover few days of our prediction periods. Also, there are few N. American sites (only Ft. Mcmurray, Bonanza creek, Missoula) that are close to major biomass burning areas, but not very good coverage of our prediction periods like other fire regions. We have added the motivation in Section 4.2.

P 16 L 30: Why do you not include the ICAP data in the intercomparison in Fig 14 as you do in Fig 13 and before?
Reply: Figure 14 is the scattering figure which can be used to compare the model results with observation. Here the blue and orange dots are good enough to include the information compared with AERONET observation. If we include ICAP comparison, there need to have the other 2 colors dots and they would be overlapped by each other and hard to get better presentation. The

correlation and RMSE of ICAP with respect to AERONET observation have been added into Table1.

P 18 L 10: Please discuss the reasons for the poorer performance of NGAC v2.
Reply: As we answered above, the NGAC v2 model is not the scope of this paper and project. Though NGACv2 is the previous global aerosols forecast system at NCEP, it has been retired and replaced by GEFS-Aerosols in the NWS since September 2020. NGACv2 has been developed by different group of scientists who are not the major developers of GEFS-Aerosols. GEFS-Aerosols and NGACv2 are two different models both in the atmospheric and chemical part. We are not the major developers of the NGACv2 and hard to conclude the reasons to cause poor performance in NGACv2 without further studies. We only got the NGACv2 history data from NCEP for comparisons. More details about the NGAC v2 model performance and evaluation can be found in Wang et al., 2018 and Bhattacharjee et al., 2018.

P 18 L 23: Please provide the resolution in km here and before of the "native" grid.
Reply: Revised.

P 18 L 28: Which resolution was used for section 4?
Reply: C384, ~25km. Added.

P 19 L 22: Please comment what the impact of the resolution on the dust emissions are. Dust emissions are known to be resolution dependent because of the respective ustar thresholds.
Reply: Yes. The dust emission is sensitive to the meteorological fields, such as surface wind the friction velocity. From our experiment in Fig.17 and Fig.19 (in previous mansucript), the impact of dust emission does not have significantly resolution dependency, which means that these meteorological fields are quite similar in these two resolutions. We have added responding comments. But according to the reviewer's suggestion in the general remarks, we have removed the discussion of C96 in the revised manuscript.

P 22 L 10: Volcanic eruptions have not been mentioned before. Please provide more details. On the other hand, one would expect that topics mentioned in the summary have been dealt with in the paper.
Reply: We have added the volcanic descriptions according to other reviewer's comments as: "Meanwhile, it is also capable of handling volcanic eruptions, which can inject vast quantities of particulates into the atmosphere. While for the predicted results in the paper, we have not included the volcanic emission into the model for the June 2019 Raikoke eruption, it may partially impact on the underprediction over high northern latitude."

P 22 L 27: Please also mention the biases against Aeronet AOD observations.
Reply: We have added the RMSE in Table 1 and corresponding discussion in Section 4 in the revised manuscript.

P 23 L 11: The paper only contains tests for different resolutions and not for different emissions in section 5.
Reply: Revised.

P 35 Fig 2: Consider introducing two separate Figures (2a = 2, 2b 3). The Fig 2b is not clear and a better caption is required.
Reply: Revised.

P 36 Fig 3: add the different reference years and global total (Tg) in the caption.
Reply: Revised. The global total (Tg) emissions have been added in Fig.3.

P 37 Fig 4: Please add the total in caption.
Reply: We have removed Fig.4 according to the reviewer's above suggestion.

P 39 Fig 6: "verified" is not the right word. You just show different plots/maps of AOD.
Reply: Revised it at "compared with"

P 40 Fig 7: Please add that you show the temporal mean of the day-1 forecasts etc.
Reply: Revised.

P 49 Fig 16: Why is NGAC not included in that Figure?
Reply: NGAC does have the OC output archived for 2016 ATOM-1 periods.

P 51 Fig 18: Please add the meaning of red and blue curve in caption.
Reply: Revised.

**References:**

Black, T., Juang, H. M. H., Yang, W. Y., and Iredell, M.: An ESMF framework for NCEP operational models, J3.1, in: 22nd Conference on Weather Analysis and Forecasting/18th Conference on Numerical Weather Prediction, Park City, UT, USA, 25–29 June 2007, American Meteorological Society, 2007.

Black, T., Juang, H. M. H., and Iredell, M.: The NOAA Environmental Modeling System at NCEP, 2A.6, Preprints, 23rd Conference on Weather Analysis and Forecasting/19th Conference on Numerical Weather Prediction, Omaga, NE, USA, 1–5 June 2009, American Meteorological Society, 2009.

Bhattacharjee, P. S., Wang, J., Lu, C.-H., and Tallapragada, V.: The implementation of NEMS GFS Aerosol Component (NGAC) Version 2.0 for global multispecies forecasting at NOAA/NCEP – Part 2: Evaluation of aerosol optical thickness, Geosci. Model Dev., 11, 2333–2351, https://doi.org/10.5194/gmd-11-2333-2018, 2018.

Wang, J., Bhattacharjee, P. S., Tallapragada, V., Lu, C.-H., Kondragunta, S., da Silva, A., Zhang, X., Chen, S.-P., Wei, S.-W., Darmenov, A. S., McQueen, J., Lee, P., Koner, P., and Harris, A.: The implementation of NEMS GFS Aerosol Component (NGAC) Version 2.0 for global multispecies forecasting at NOAA/NCEP – Part 1: Model descriptions, Geosci. Model Dev., 11, 2315–2332, https://doi.org/10.5194/gmd-11-2315-2018, 2018.

---

## Author Comment (AC2)

*Reply to anonymous referee #2*

Review of "Development and Evaluation of the Aerosol Forecast Member in NCEP's Global Ensemble Forecast System (GEFS-Aerosols v1)" by Zhang et al. for publication in Geoscientific Model Development
The paper presents a description of the new GEFS-Aerosols modeling capability that is part of the FV3-based ensemble forecasts of the Global Forecast System (GFS). A number of experiments are performed with this system, and results are explicitly shown evaluating different biomass burning emissions assumptions and impacts of model horizontal resolution. Model results are compared to MODIS and VIIRS observations, AERONET and ATom data, and results from the GEOS-FP, MERRA-2, ICAP, and NGACv2 model-derived products. The model is shown to have considerably better performance relative to its predecessor NGACv2 system when compared to data sets and independent model products. Residual differences in the GEFS-Aerosols performance versus observations and models are speculated at.
The paper is overall well organized and the figures are for the most part clear (I detail some places below where I have suggestions to improve). I recognize here this is a significant update to the modeling capabilities for this major meteorological forecasting system, and I appreciate the progress the authors are making on this work. I nevertheless have a number of concerns about the paper as prepared here that I wish to see addressed before it can be published in a final form. I have many minor suggestions articulated below, but I here will lay out a few more major points.
Reply: We really appreciate the reviewer's very helpful comments and suggestions. The paper has been revised throughout based on all the general and specific comments listed by the reviewers, including the text, figures, references, etc.

First, the model description is lacking in some significant respects. In particular, there is no description of loss processes in the aerosol scheme and how they impact the simulation. This is unfortunate because in a number of places it is asserted that uncertainties in wet removal schemes explain differences between the model and observations. A general description of the approach would be helpful here, and it would be useful also to see differences in the large-scale and convective-scale precipitation between the different resolution runs as a means to explore these differences.
Reply: We thank the reviewer's suggestion. We have added the model descriptions about sink and source processes, including the convective wet scavenging, large scale wet removal, dry deposition etc. in Section 2.1.1 and Section 2.1.2. According to the other reviewer's suggestion, we have removed the discussion and comparison of different resolution in Section 5, so we did not include the comparison of precipitation between different resolution.

More generally, a budget analysis for a new modeling system is a useful add (see e.g., Textor et al. 2006, www.atmos-chem-phys.net/6/1777/2006/) for some inspiration. It is helpful to see how the lifetime of your model is similar to and different than other systems.
Reply: We thank the reviewer's suggestion. We have the other group leading by Li Pan is working on the budget analysis and aerosol lifetime in GEFS-Aerosols. They just finished the draft and plan to submit it soon:
Pan, L., P. S. Bhattacharjee, L. Zhang, R. Montuoro, B. Baker, J. McQueen, G. A. Grell, S. A. McKeen, S. Kondragunta, X. Zhang, G. J. Frost, F. Yang, I. Stajner: Analyzing GEFS-Aerosols

annual budget to understand simulated BC, OC, Dust, Sea salt and Sulfate results in the model, to be submitted, 2022.

Second, the comparisons between the GEFS-Aerosol simulation and the comparison datasets is in most cases only qualitative. There are any number of places where the performance is described as "very good" or "better" than this or that. For the most part these are not very helpful qualifiers, and in some cases I can't reconcile the assertions with the graphics presented, or at least I don't know what exactly is being highlighted. Better is something like the presentation in Figures 10 and Table 2, which are at least quantitative (well, semi-quantitative in Figure 10). These provide more objective measures of quality. Please address this in the revisions.

Reply: We have revised the manuscript throughout with more quantitative statements and descriptions in the evaluation instead of using "very good" or "better". We also added the correlation and the RMSE values for GEFS-Aerosols, ICAP and NGACv2 with respect to AERONET observation in Table 1. Our descriptions and discussions have been modified to include these statistical results in Fig. 9, Fig. 10, Fig. 11, and Fig. 12 in the revised manuscript.

Third, and related, where discrepancies within the comparisons are noted there are appeals to wet removal schemes, plume rise model, dust emissions, and the like. Mostly these assertions are not grounded in anything presented in the paper. A compositional analysis that links underestimates in Europe to Saharan dust emissions (is that really the culprit?) would be helpful. Something similar (sensitivity tests?) to the points about wet removal too. I note a reference below that is relevant, but in particular it is pretty clear that this model suffers somewhat from a common problem in aerosol models with insufficient scavenging of especially black carbon in convective updrafts. Further expansion on this point should be included.

Reply: We have revised the descriptions and assertions. About the European AOD underprediction, this was a mistake in previous descriptions, it is not related to dust, however the sulfate AOD. We have modified it. Also, in several places about the assumptions related to wet removal, we have revised them and emphasized that it needs further investigations. We also added the references related to the black carbon wash out issues that also indicated in other models.

Finally, also noted below, the authors have chosen to evaluate the model performance with a focus on a perturbed period following the June 2019 Raikoke eruption. I note there is no indication of whether the model includes volcanic emissions at all, and Raikoke is evidently not in the simulation. If other pre-COVID periods were available for this evaluation I would prefer that, but at the least I think some acknowledgement of this state would be important to introduce as a caveat, probably most relevant to discussion of high northern latitude biomass burning.

Reply: The model has the capability to include the volcanic emission for $SO_2$ which is based on the estimate of injection height and eruption time. While the prediction results in the paper, we have not included the volcanic emission into the model for the 2019 Raikoke eruption. We have emphasized it in the revised manuscript as "While for the predicted results in the paper, we have not included the volcanic emission into the model for the June 2019 Raikoke eruption, it may partially impact on the underprediction over high northern latitude.".

For the pre-COVID periods, other than 2019, we only the ATOM-1 evaluation is based on 2016 summer. Because the GBBPEx operational data for GEFS-Aerosols model only launched from the 2019 summer for operational prediction other than the ATOM-1 periods, so we can not run the

retrospective experiments before 2019 summer except the ATOM-1 periods. We have added some acknowledgements about the Raikoke eruption in Section 6.

Reply: Revised

Page 4, Line 26: I don't see it explicitly, but I presume in the GEFS-Aerosols member the aerosols are not in any way interactive with the radiation, clouds, etc. Please clarify that's the case. Also, assuming so, how does GEFS-Aerosols differ from other GEFS members except for the prognostic aerosols? Is it meteorologically equivalent to another member of the ensemble?
Reply: Yes. The aerosol feedback has not been included there are not in any way interactive with the radiation, clouds. We have clarified it. The only differences of between the GEFS-Aerosols other GEFS members is the prognostic aerosols, and the meteorological parts of GEFS-Aerosols are the same to other ensemble members. We have added these descriptions in the revised manuscript in Section 2.1.3 and Section 6.

Page 5, Line 11: Citations for FV3? I think it has quite a literature.
Reply: Cited.

Page 5, Line 17: I have no context to understand what GFSv15 and GEFSv12 mean. Please clarify.
Reply: The atmospheric model of FV3GFS include the dynamical core of FV3 and physical scheme of GFS scheme, here the GFSv15 is the version of physical scheme, and GEFSv12 is the version of FV3GFS with ensemble members. We have clarified it in Section 2.1.1.

Page 5, Line 17: Here or somewhere nearby it would be relevant to state the model resolution of your simulations, including also the vertical coordinate. The horizontal resolution is referred to finally in the paper much later, but I don't see the vertical resolution discussed at all.
Reply: We have added the model vertical resolution descriptions in Section 1 and Section 2.

Page 5, Line 25: Please clarify if you are in fact getting DMS emissions from Lana et al. (2011). If so, that is a departure from GOCART, which uses the DMS seawater concentrations and then determines emissions dynamically based on surface wind speeds. If using DMS direct from Lana et al. (2011), for what year and seasonal variability are you assuming?
Reply: We thank the reviewer's very good comments. We have modified it as "The marine dimethyl sulfide (DMS) emission is calculated as a product of sea water DMS concentration and sea-to-air transfer velocity as described by Chin et al., [2000]."

Page 5, Line 27: In the abstract you refer to a HRRR-based plume rise model, but here you say WRF-Chem. On page 11 you refer again to HRRR-based model heritage from the WRF-Chem. Please clarify this consistently throughout.
Reply: Yes, the plume-rise module in HRRR-Smoke is originally from WRF-Chem, with some modifications and application of the FRP in HRRR-Smoke. We have clarified it.

Page 6, Line 2: The GOCART model referred to here with the 5-bin sea salt is in Colarco et al. (2010), doi: 10.1029/2009jd012820

Reply: We thank the reviewer's very good suggestion. We have updated the citation.

Page 6, Line 6: The "S" and "A" terms are not obviously defined in the text. I cannot find a reference for the FENGSHA scheme here, or at least the Tong et al. 2017 citation is missing in the references. Please state what "S" and "A" are (where they derive from) and add the citation.
Reply: We have the descriptions of "S" and "A" terms as "$A$ represents particle supply limitation (availability), $\rho$ is air density, $g$ is gravitational acceleration, $S$ is the soil erodibility potential". Also, the citation of Tong et al., 2017 has been added: "Tong, D. Q., Wang, J. X. L., Gill, T. E., Lei, H., and Wang, B. (2017), Intensified dust storm activity and Valley fever infection in the southwestern United States, *Geophys. Res. Lett.*, 44, 4304– 4312, doi:10.1002/2017GL073524."

Page 7, Line 1: Later in the text wet removal is appealed to in various places to explain the agreement (or lack thereof) with ATom data. I note there is no mention of loss processes and how parameterized in the model. Are the loss processes also in the same sequence as the emissions in GEFS-Aerosols? What is the process order?
Reply: We have added descriptions about wet removal and other chemical processes about source and loss in Section 2.1.2 as "The metrological fields (such as land use and other climatological surface fields, vegetation type etc.) are imported from the FV3 atmospheric model to the chemical model to drive the aerosols components. They are consistent in the FV3 atmospheric model and chemical model. Other than the aerosols convective wet scavenging, all the chemical related processes of source and sink, such as emission, dry deposition, settling, large-scale wet deposition, chemical reactions are handled by the chemical model. The large-scale wet deposition and dry deposition modules are from WRF-Chem for GOCART aerosols scheme, which are column model driven by meteorological input from atmospheric model. Large-scale wet removal of aerosols includes below-cloud removal (washout) following Easter et al. [2004] and the details of below-cloud wet scavenging via interception and impaction can be found in Slinn [1984]. The dry deposition is the same as Chin et al. [2002]. After updating the chemical tracers in chemical model, they are passed back to FV3 atmospheric model for transport and advection."
We also described the chemical sequences in Section 2.1.3 as "All aerosol composition and emission-related processes are computed in GEFS-Aerosols after the atmospheric physics has been advanced and passed to the chemical model following the sequences as emission, settling of dust and sea salt, plume-rise of fire emission, dry deposition, large-scale wet deposition, chemical reactions and carbonaceous aerosol updating."

Page 7, Lines 12 - 16: This text just reads out of place here as it is descriptive of the GEFS configuration and not the aerosols themselves. This belongs I think in Section 2.1.1.
Reply: We have moved this part to Section 2.1.1

Page 7, Lines 22-23: "aerosol optical properties from NASA" is not terribly descriptive. If from GEOS/GOCART please cite appropriate sources (e.g., Colarco et al. 2010, Colarco et al. 2014).
Reply: Revised by adding the citations.

Page 8, Line 10: MERRA-2 do not provide forecasts, or anyway not in some form readily accessible. It is a reanalysis and I suspect you are looking at those products, which might just be described as state snapshots or averages.
Reply: Revised is as "AOD product". Thanks for the comment.

Page 8, Line 32: The GEOS system I think referred and used here is the near-real time GEOS-Forward Processing (GEOS-FP) system. Suggest that terminology. And my understanding is the "branding" is no longer using "GEOS-5" but simply "GEOS".
Reply: According to the other review's suggestion. We have removed the comparison with GEOS-5 and all use the MERRA-2 product. Thanks for the review's comment all the same.

Page 10, Line 6: What is the spatial resolution of the CEDS inventory used here? And are you in fact using the earlier CEDS inventory cited here and not more recently available versions that go through 2019?
Reply: The original spatial resolution of CEDS 2014 is 0.5x0.5 degree. When we started the real-time run from 2019 summer, at that time, the most recent CEDS emission is 2014 version. The 2019 version CEDS is available since 2021. The CEDS 2019 will be updated in our next operational updating this summer, which is still under evaluation now.

Page 10, Line 19: What does "GOCART model background fields" refer to? I infer the oxidants. Please clarify. "Does "NASA GEOS/GMI" model refer to the MERRA-2 GMI version (https://acd-ext.gsfc.nasa.gov/Projects/GEOSCCM/MERRA2GMI/), or something else?
Reply: Yes. The "GOCART model background fields" refer to $H_2O_2$, OH and $NO_3$. We have rewritten this part and moved to Section 2.1.2. The GMI model link has been added. Thanks for the reviewer's information.

Page 10, Line 23: I find this description confusing and am not sure what is being described versus shown in Figure 4. GBBEPx is stated to blend emissions from several sources…is that really what it is doing, is blending QFED with other emission sources? QFED I think would not be referred to as "MODIS QFED" like here as it is not a MODIS product, but derived from MODIS observations. Second is a reference to 3BEM emissions which is merged with WF_ABBA. But Figure 4 calls this "MODIS" which I don't understand. Finally, the plume rise model is mentioned to take input from FRP data. How does this relate to either of the emission products mentioned here?
Reply: According to the other reviewer's suggestion, we have removed this part and Figure 4-5. But we would also like to answer the above questions. We did not use the QFED fire emission. The Experiment 1 is using an online fire emission module in PREP-CHEM-SRC tool, which need the input data of MODIS and WF_ABBA fire product to calculate the aerosol emission based on different aerosol scheme, including GOCART scheme. It also includes a simple vertical profile to redistribute the fire emission vertically. The Experiment 2 is only using the GBBEPx fire emission at the surface. But it is not based on any observation data. The Experiment 3 is using the GBBEPx fire emission and FRP product as the input for the plume-rise module from HRRR-smoke to generate 3D fire emission. The Experiment 3 is the fire configuration in the operational GEFS-Aerosols.

Page 11, Line 13: It is really hard to read Figure 4, even blown up on a screen, in relation to the comments made about it. I can clearly see more fire spots across the northern latitudes in the GBBPEx emissions, but I cannot tell if the magnitudes are different or not in general because the points are too small to see. It is certainly not evident that emissions are greater in southern Africa (Line 15). My suggestion would be to show a temporal average (a month, a season) to make this point, and you can numerically refer to the relative number of fires observed if you need to.

Reply: According to the other reviewer's suggestion, we have removed the discussion about Experiment 1 and Experiment 2, also removed corresponding figures of Figure 4 and Figure 5.

Reply: According to the other reviewer's suggestion, we have removed the discussion about Experiment 1 and Experiment 2, also removed corresponding figures of Figure 4 and Figure 5. But we would like to answer the above questions. The Experiment 1 is using an online fire emission module in PREP-CHEM-SRC tool, which need the input data of MODIS and WF_ABBA fire product to calculate the aerosol emission based on different aerosol scheme, including GOCART scheme. It also includes a simple vertical profile to redistribute the fire emission vertically. But it is not based on any observation data. The Experiment 3 is using the GBBEPx fire emission and FRP product as the input for the plume-rise module from HRRR-smoke to generate 3D fire emission. The Experiment 3 is the fire configuration in the operational GEFS-Aerosols.

Reply: Only four models are used to compute ensemble mean in ICAP for total AOD calculations (NASA,NRL, JMA and ECMWF). NGAC is not used for total AOD in ICAP. So, NGAC is not withheld from ICAP in the total AOD comparison. We have clarified it in Section 2.2.3.

Reply: Yes. But according to the other reviewer's suggestion, we have removed the part related to this discussion in Section 3.3 and Figure 4-5.

Reply: According to the other reviewer's suggestion, we have removed this part in Section 3.3 and Figure 4-5. We have used more statistics values to describe the model performance.

Reply: We have revised it throughout the manuscript.

Reply: We thank the reviewer's suggestion. According to the other reviewer's suggestion, we have removed this part in Section 3.3 and Figure 4 and Figure 5.

I also want to point out here (and later in relation to Figure 7) that you have chosen an interesting period for analysis owing to the June 22, 2019, eruption of Raikoke in the northwest Pacific, which was a significant perturbation to the high latitude aerosol environment. There is no mention of volcanic emissions in GEFS-Aerosols until the conclusions where it seems like a future extension, so I presume Raikoke is omitted from the analysis. Likely the ICAP models (and for sure MERRA-

2) do not explicitly account for the eruption, but they could catch some aspect of it through aerosol data assimilation. This needs to be noted somewhere in the discussion. Look especially at the high latitude MODIS points in Figure 7.

Reply: We would like to thank the reviewer for bringing this up. In GEFS-Aerosol, though it has the capability to include the volcanic emission during eruption time, but for the operational and real-time forecast, which need to catch up the real date, it does not include the volcanic emission. Because we can not predict when the volcanic eruption would happen. We have emphasized it in Section 2 and Section 6 in the revised manuscript. Also, we have looked into individual figures of MODIS from July 5 to November 30, 2019 (the period used in the Figure 6 in the revised manuscript). Though the AOD over high latitude is available in MODIS during this time period, the enhanced AOD may also come from Siberian fire, long-range transport from other sources, or from volcanic emission. We have added the discussion into the summary as "While for the predicted results in the paper, we have not included the volcanic emission into the model for the June 2019 Raikoke eruption, it may partially impact on the underprediction over high northern latitude."

Page 13, Line 22: Please clarify here and elsewhere what we're looking at and how it's done. Figure 8 refers to Day 1 AOD forecast biases. I *think* what you are doing is running a 1 day forecast of the aerosol and then resetting the meteorology to the new analysis and making another 1 day forecast, and so on. So you are showing in Figure 8 the ~4 month mean of those 1 day forecast outcomes? How is that compared to the GEOS analyses mentioned here? Are you also looking at GEOS forecast outputs? Or the analysis itself? Are they compatible with what you are doing? Does it matter? Is this just a simple difference of the multi-month means?

Reply: We compared each day model forecast hours with same day GEOS5 analysis or other or reanalysis data (we have used only FP GEOS5 analysis, not GEOS5 forecast) and computed the AOD statistics (Bias, RMSE, correlation etc.) for each grid for each pair of model and analysis for that model forecast hour. We then computed that for the entire 4 months of the study period and averaged it over the entire 4 months for each grid points. We have use DTC MET Tool to calculate these statistical values, this method gives an overall estimate of systematic bias of the model in spatial and temporal scale. We have clarified it in Section 2.2, similar way is also applied to the MERRA-2.

Page 14, Line 1: How might you expect emissions to differ in the 2019 simulation years versus the 2014 valid year for the CEDS inventory used here?

Reply: We would expect some overpredictions over China by using the CEDS 2014 emission for the 2019 simulation. It well known that strong actions have been taken to improve the worsening atmospheric environment in the last 10 years in China, including cutting down the pollutant emissions with forced installation of catalytic converters on vehicles, building clean-coal power generation systems, prohibiting open burning of crop, etc. (Chen et al., 2017; Zhang et al., 2012; Liu et al., 2016). Currently, the $PM_{2.5}$ pollution occurrence has reduced to meet the goals in the Air Pollution Prevention and Control Action Plan (issued by the State Council of China, http://www.gov.cn/zwgk/2013-09/12/content_2486773.htm). Considering the decreasing emission trend over China, the CEDS 2014 anthropogenic would result in the overprediction in 2019. We have added this discussion in Section 3.1 as "It should be noted that these anthropogenic emissions data are not impossible to catch up the date of real-time forecast. And it normally has time lag and represents the emissions of a different previous years. The inconsistency may have

some impact on the predictions in 2019. But that is the most recently available version of anthropogenic emission. It well known that strong actions have been taken to improve the worsening atmospheric environment and decrease the emission over China in the last 10 years (Chen et al., 2017; Zhang et al., 2012; Liu et al., 2016). Considering the decreasing emission trend over China, the CEDS 2014 anthropogenic would result in some overprediction after 2014."

Page 14, Line 28: Suggest adding some statistics of the comparisons tabularly in Table 1. It is hard to read the colors in Figure 10 quantitatively.
Reply: Thanks to the reviewer. We have provided correlation and RMSE statistical values for all the 60 sites (including the one shown in previous Figure 10 in tabular) in Table 1, also added the RMSE figures in Fig.5 in the revised manuscript.

Page 15, Line 20: I don't see what you are referring to here, and if anything ICAP looks closer to the AERONET points in Figure 11b at the time indicated.
Reply: Revised. "GEFS-Aerosols is able to predict the two AOD enhancements in mid-October and early November, which is quite comparable as ICAP. The correlation (RMSE) is 0.856 (0.15) and 0.936 (0.09) for GEFS-Aerosols and ICAP with respect to AERONET at the site of Itajuba, only 0.451 (0.22) for NGACv2."

Page 15, Line 32: I'm not sure what is meant by saying GEFS is both comparable to but slightly lower than ICAP.
Reply: Revised as "the GEFS-Aerosols prediction is slightly lower than ICAP by about 5-10%."

Page 16, Line 7: Only seven sites are shown.
Reply: Revised.

Figure 14: What is the shading shown on the map in the top left corner?
Reply: That is dust AOD on a specific day which is not necessary to show, we have modified the figure to only show the location of the sites.

Page 17, Line 25: There is nothing here that supports the assertion that a low bias over Europe is caused by underestimates in dust emissions. Please explain further.
Reply: Revised. It was a mistake in the previous description which is not related to dust emission. We have modified it as "The large absolute low biases from August to early October 2019 and March 2020 in Europe which are associated with GEFS-Aerosols underestimates of sulfate AOD (Fig. 8)".

Page 18, Line 6: There is nothing that supports the statement that under predictions are due to errors in emission or wet removal processes. Or, put differently, this statement doesn't really explain anything in the nature of the figure comparison shown.
Reply: Thanks for the comments. Yes. The biases need further investigation. We have revised it as "Both the Eastern and Western US regions exhibit GEFS-Aerosols low biases of about 5-30%, with the largest differences in Eastern US occurring in August 2019. However, the trends of total AOD temporal variations with low in summer and high in winter in the GEFS-Aerosols prediction and the MERRA-2 reanalysis are quite consistent over Eastern and Western US. The minor under predictions by GEFS-Aerosols need further investigation."

Page 18, Line 24: I think I know what "log-Z AGL" means, but please explain.
Reply: Explained.

Page 19, Line 28: I note that Figure 18 does not include any labeling Pacific versus Atlantic. Further, this is a very challenging figure to read without blowing up quite large on the monitor. I suggest you split into one figure for each species to allow more room. Finally, what you call in the text here "bias" is presented in the figure as a ratio. Please use consistent terminology.
Reply: Revised. We have added the label of Pacific and Atlantic in Fig. 16 and Fig. 17 in the revised manuscript.

Page 20, Line 3: Not sure about "all of the three experiments." I see two experiments.
Reply: Revised.

Page 20, Line 29: How do you quantify "very good" performance?
Reply: Revised. "The model results show similar pattern as the ATOM-1 in reproducing the profiles of OC even using log scale".

Page 20, Line 32: I don't follow that the model is able to capture variations in the latitude-height profiles. Figure 18 shows that BC is overestimated in the models by a factor of 10 at higher altitudes. This is by the way a known problem in many models that they do not adequately scavenge BC (see e.g., Wang et al. 2014, doi: 10.1002/2013JD020824 and later).
Reply: We thank the reviewer's very good comments. We have revised it. "Overall, predicted BC (middle column of Figure 20) is able to capture the decreasing trend with increasing altitude in the latitude-height profiles, however they are underpredicted in the biomass burning plumes near the tropics from the surface to 5 km height in both model experiments, which have been seen in other models due to insufficient scavenge (Wang et al., 2014, Choi et al., 2020)."

Page 21, Line 4: Here injection height and scavenging are again appealed to for explanations for differences. What is the impact of the injection height parameterization, and how is that evaluated in this model?
Reply: The injection height parameterization is based on the plume-rise module in Grell et al., 2011, a 1-D, time-dependent cloud model with predicted Met. Fields were used online to calculate the injection heights as well as the vertical distribution of the emission rates. So the injection heights is based on dynamical calculation of weather conditions. The mainly impact of applying the plume-rise module it that the fire emission is 3-D distribution based on the online calculating injection height. But this injection height is difficult to evaluated because there is no accurate observation to compare. What we may validate is the vertical profile of the aerosols from fires. We thank the reviewer' suggestion, we have modified the descriptions as "It appears the model does not reproduce the enhancements of BC at 1-4 km height very well over this area. It may be possibly due to relative weak convection or a low modeled injection height that the fire emission has not been lifted enough to this altitude, which need further studies."

**References:**

Chen, J. M., Li, C. L., Ristovski, Z., Milic, A., Gu, Y. T., Islam, M. S., Wang, S. X., Hao, J. M., Zhang, H. F., He, C. R., Guo, H., Fu, H. B., Miljevic, B., Morawska, L., Thai, P., Fat, L., Pereira, G., Ding, A. J., Huang, X., and Dumka, U. C.: A review of biomass burning: Emissions and impacts on air quality, health and climate in China, Sci. Total Environ., 579, 1000–1034, https://doi.org/10.1016/j.scitotenv.2016.11.025, 2017.

Grell, G., Freitas, S. R., Stuefer, M., and Fast, J.: Inclusion of biomass burning in WRF-Chem: impact of wildfires on weather forecasts, Atmos. Chem. Phys., 11, 5289–5303, https://doi.org/10.5194/acp-11-5289-2011, 2011.

Liu, F., Zhang, Q., van der A, R. J., Zheng, B., Tong, D., Yan, L., Zheng, Y., and He, K.: Recent reduction in $NO_x$ emissions over China: synthesis of satellite observations and emission inventories, Environ. Res. Lett., 11, 114002, https://doi.org/10.1088/1748-9326/11/11/114002, 2016.

Zhang, Q., He, K., and Huo, H.: Cleaning China's air, Nature, 484, 161–162, https://doi.org/10.1038/484161a, 2012.

---

## Author Response (AR2)

**Reply to anonymous referee #1**

**Overview**

The response to my review was often not clear enough about the actual changes to the new version of the manuscript. That made it often difficult to check if the modifications of the manuscript addressed the comments. The main concern of the first review was that the initial manuscript was too complex: various model versions and data sets were used in an often in-consistent way. The revised manuscript has improved on this aspect, but several modifications are still required before it can be recommended for publication.

**Reply:** We thank the reviewer's very helpful comments and suggestions. The revised manuscript has been revised again taken into account the reviewer's comments as following. We listed all the changes one by one as followings.

Specific comments:
I had asked in my first review to provide more information about the reasons for the differences between GEFS and NGACv2. In the response the authors say that is difficult to specify and refer to the complexity of the different atmospheric modelling approaches. But the manuscript was not updated accordingly. If the authors believe most of the changes are dues to the meteorological modelling they should say so and try to give evidence as good as possible. For example, wet deposition differences as suggested by the other reviewer, are a good candidate. Some of the improvements seems to come from the different emissions, especially for fires. The authors should mention this too in the conclusion section. It is understood that an exact attribution of the causes of model changes is difficult, but the authors should not avoid that important topic completely.
**Reply:** We thank the reviewer's very good suggestions. We have mentioned these important differences between GEFS-Aerosols and NGACv2 that may contribute to the improvements for GEFS-Aerosols model in the revised manuscript, corresponding discussions are also added into conclusion section. Please see them at: P13: L21-23; P18: L8-9; P22: L30-P23: L1; P23: L6-8; P24: L5-7; P24: L15-19.

The table 2 is a welcome addition but further details and clarification are required: Was the same physics package used (GFSv2015 vs GFSv15 - or is that a typo), which emissions are used for NGACv2 in the result shown in the paper (QFED v2 or GBBEPx) Is the exactly the same GOCART version used for non-dust aerosol etc.
**Reply:** This is not a typo, GFSv15 was implemented in 2019, while GFSv2015 was implemented in 2015 (corresponding version should be GFSv12, but it did not call GFSv12 at that time). NCEP name the physics package using GFSv15, v16… since the GFS package was implemented into FV3 from 2019. We have revised clearly in **Table 3**.

In 2.2.3 (P11: L6-10), we have described the fire emission used in NGACv2 as "The fire Emissions of carbonaceous aerosols and SO2 are from Global Biomass Burning Emission Product-extended [GBBEPx, Zhang et al. 2014]. GBBEPx emissions are blended from NESDIS's Global Biomass Burning Emission Product from a constellation of geostationary satellites [GBBEP; Zhang et al., 2012] and GMAO's Quick Fire Emissions Data Version 2 from polar-orbiting satellites [QFED2; Darmenov and da Silva, 2015]". We also have clarified them in **Table 2**.

Yes. The GOCART version is not exactly the same version for non-dust aerosol, because one is based on GOCART in WRF-Chem, while the other one is based on the GOCART within GEOS-5. We have clarified them in **Table 2**.

The new manuscript is improved as it focuses more on the comparison of the "new" system "GEFS-Aerosols v1" with the "former" system "NGACv2". However, the nomenclature is still inconsistent. The authors should carefully check text, figures, tables and captions to use always the same short name for the referred to model versions. All the occurrences, also in the figures itself need to be corrected. For example "NGAC" , "NGAC-v2" . NGAC-GOCART" is used in the text and figures. The GEFS aerosol is in some places referred to as "FV3-C384" (Table 3) or GBBEPx (e.g Figure of 18).

**Reply:** We thank the review's very good suggestions. We have revised it throughout the manuscript, including the tables and figures to use the uniform names.

As pointed out in my review, the authors should clarify that the paper is not about interactive aerosol in the radiation scheme. Still almost half of the introduction section is dedicated to that topic, which is not discussed in the paper. That is misleading and the section should be removed or substantially shortened.

**Reply:** We really appreciate the reviewer's very helpful comments. We have removed most of this part and shortened it substantially, see P3 L10 to P3 L19.

On the other hand, global aerosol forecasting systems, of which GEFS Aerosol is another example are not sufficiently discussed in the introduction section. For example, the global aerosol forecasting system run by ECMWF/CAMS, JMA, NASA GSFC/GMAO and NRL/FNMOCCAMS or other system participating in the ICAP effort should be discussed and current challenges of aerosol forecasting could be reflected on.

**Reply:** We really appreciate the reviewer's very helpful comments. We have added other global aerosol forecasting systems, including the ICAP, into the introduction. See P3 L20 to P4 L12.

The figure captions and legends still require a careful review. For example, Figure 16 and 17 do not include the information what is shown. In Figures 18 and 19 it is not clear which species and dates are shown in which panel. All panels have the time label 8/15/16 but the captions say the plots are shown for two dates. Why is only NGACv2 MM shown and not for dust, OC, BC and SO2.

**Reply:** We have revised the figures from Fig.14 to Fig. 19. We added more clear titles into the figures to show the species, we also revised the figure captions to make them clearly.

For NGACv2 results, there is only the dust concentration being archived in the data storage for 2016 at NCEP, the concentration of other aerosol species are not available for 2016.

**Reply to anonymous referee #2**

**Overview**

I appreciate the effort the authors took in revising the manuscript, and in light of the reviewer 1 comments that necessitated a significant restructuring and reorganizing of the paper I find most of my remaining comments are satisfactorily addressed. I especially appreciate the inclusion of statistics in Table 1 and addition of Tables 2 and 3 that further bolster the results with quantification instead of qualification. The paper could use a thorough read again for grammar, but mostly I only have minor comments below and find the paper suitable for publication pending addressing those.
Reply: We thank the reviewer's very helpful comments and suggestions. The revised manuscript has been revised again taken into account the reviewer's comments as following.

**Minor comments**

Figure 1: What is "GSDCHEM"?
**Reply:** Revised as "Diagram showing the components within the NEMS infrastructure".

Page 5, Line 13: Don't understand "no execution time" sentence, awkwardly constructed and unclear what is meant.
**Reply:** In the first review comments, the other reviewer raised a question as "Please indicate the resolution of the 31 non-aerosol members and the resolution of the aerosol member. Are they the same? How does the potentially increased cost and execution time of the aerosol member impact the execution time of the ensemble as a whole?" Then we replied to it as "The resolution of the 31 non-aerosols members and the aerosols member are the same. The aerosol member only performs 120 hours forecast, however, the non-aerosol members perform 384 hours forecast, so the no execution time issue because the aerosol member finishes the forecast before other non-aerosol members, so there is no execution time issue."
We have modified it as "In the operation, there is no execution time by including the aerosols component as one of the ensemble members since this member only performs 120 hours forecast by including aerosol component, which is shorter than other members without aerosol component that perform 384 hours forecast."

Please re-read 2.1.1 carefully to make sure it is clear and logical. It seems a bit jumbled.
**Reply:** We have rewritten and reorganized the section 2.1.1 as "The global Finite-Volume cubed-sphere dynamical core (FV3) developed by GFDL was chosen by NOAA as the non-hydrostatic dynamical core to be the Next Generation Global Prediction System (NGGPS) of the National Weather Service in the US [Black et al., 2021]. Currently, the FV3 was successfully implemented within the physical scheme of GFS version 15 (named as FV3GFS v15), which became operational on June 2019. It has the capability to provide the metrological basis for coupling with aerosol prediction component. The GEFS is a weather forecast modeling system made up of 31 separate forecasts, or ensemble members, which have the same horizontal (~25 km) and vertical resolution (64 layers from the surface to 0.2 hPa). The GEFS-Aerosols model is only using one of the same weather model as other GEFS members except it includes the prognostic aerosols from the coupling aerosol component. The NCEP started the GEFS to address the nature of uncertainty in weather observations that are used to initialize weather forecast models and uncertainties in model representation of atmospheric dynamics and physics. The aerosol component coupled with

FV3GFS v15 has been merged into the GEFS, as a single ensemble member named as GEFS-Aerosols, for real-time and retrospective forecast that preceded operational implementation, which occurred in September 2020.

In GFS v15, all sub-grid scale transport and convective deposition related to aerosol is handled inside the atmospheric physics routines of simplified Arakawa–Schubert (SAS) scheme. It requires consistent implementation of positive definite tracer transport and wet scavenging in the physics parameterizations, which was implemented subsequent in the forecast system of GEFSv12."

Page 6, Line 20: "capability" instead of "capable"
**Reply:** Revised.

Page 8, lines 14-16: This sentence indicates a significant change to the values from the optical LUTs, but I have no idea what they actually did. The model is not including nitrate, so they are scaling things up. How is that decided when/where to do that?
Reply: There was a bug in the AOD calculation before when applied the optical LUTs before, we found it and fixed it, there is not necessary to use any scaling and it has been cancelled now. In our next generation forecast system using UFS-Aerosols, the nitrate will be included.

Page 11, Line 21: "It is well known"
Reply: Revised.

Page 12, Line 8: I don't know what "two different fire emission datasets" are referred to here. GBBEPx v3 appears to be what is used here, which is merging several things together. I don't follow if this is differenced from something else somewhere.
Reply: Revised it as "The 1-D cloud module is able to be applied GBBEPx v3 fire emissions datasets to account for plume-rise that distributes the fire emissions vertically and better simulate the fire events and pollution transport of smoke plumes."

Page 14, Line 1: "more comparable to the MODIS observation" than what?
Reply: Revised it as "The high AOD over southern Africa and northern India is more comparable to the MODIS observation that of NGACv2".

Page 14, Line 33: Visual inspection of Figure 8 suggests that both models biases over the Taklimakan Desert are much larger than the 0.1 bias indicated.
Reply: Revised it as "Both GEFS-Aerosols and NGACv2 total AOD have small negative biases (~0.3-0.5) relative to MERRA-2 over the northwestern China dust source region."

Page 16, Line 16: Correlation at Maun Tower is lower in GEFS than for ICAP, per Table 1.
Reply: Revised it as "The correlation coefficients at the sites of Ascension Island and Lubango are much high than those of ICAP (see Table 1)"

Figure 12: I'm surprised to see the correlations for NGAC so poor at Cape Verde and Tenerife. This is large scale dynamics and not so strongly coupled to source processes. Are they using realistic meteorology at all?
Reply: Yes. They are using real-time forecast meteorological data.

Page 19, Line 16: Looks to me like the model "underestimates" rather than "overestimates"
Reply: Revised as "underestimates".

Figure 14, 15, 18, and 19 would all benefit from titles at the top that explain the quantity shown rather than referring to instrumentation jargon in the legend (e.g., SP2ngm3 in Figure 18).
Reply: Revised.